# On the Decision Boundaries of Neural Networks. A Tropical Geometry Perspective

## Abstract

This work tackles the problem of characterizing and understanding the decision boundaries of neural networks with piecewise linear non-linearity activations. We use tropical geometry, a new development in the area of algebraic geometry, to characterize the decision boundaries of a simple network of the form (Affine, ReLU, Affine). Our main finding is that the decision boundaries are a subset of a tropical hypersurface, which is intimately related to a polytope formed by the convex hull of two zonotopes. The generators of these zonotopes are functions of the network parameters. This geometric characterization provides new perspectives to three tasks. **(i)** We propose a new tropical perspective to the lottery ticket hypothesis, where we view the effect of different initializations on the tropical geometric representation of a network's decision boundaries. **(ii)** Moreover, we propose new tropical based optimization reformulations that directly influence the decision boundaries of the network for the task of network pruning. **(iii)** At last, we briefly discuss the reformulation of the generation of adversarial attacks in a tropical sense, where we elaborate on this in detail in the supplementary material.[1]

## 1 Introduction

Deep Neural Networks (DNNs) have demonstrated outstanding performance across a variety of research domains, including computer vision (Krizhevsky et al., 2012), speech recognition (Hinton et al., 2012), natural language processing (Bahdanau et al., 2015; Devlin et al., 2018), quantum chemistry Schütt et al. (2017), and healthcare (Ardila et al., 2019; Zhou et al., 2019) to name a few (LeCun et al., 2015). Nevertheless, a rigorous interpretation of their success remains elusive (Shalev-Shwartz & Ben-David, 2014). For instance, in an attempt to uncover the expressive power of DNNs, the work of Montufar et al. (2014) studied the complexity of functions computable by DNNs that have piecewise linear activations. They derived a lower bound on the maximum number of linear regions. Several other works have followed to improve such estimates under certain assumptions (Arora et al., 2018). In addition, and in attempt to understand some of the subtle behaviours DNNs exhibit, *e.g.* the sensitive reaction of DNNs to small input perturbations, several works directly investigated the decision boundaries induced by a DNN for classification. The work of Moosavi-Dezfooli et al. (2019) showed that the smoothness of these decision boundaries and their curvature can play a vital role in network robustness. Moreover, the expressiveness of these decision boundaries at perturbed inputs was studied in He et al. (2018), where it was shown that these boundaries do not resemble the boundaries around benign inputs. The work of Li et al. (2018) showed that under certain assumptions, the decision boundaries of the last fully connected layer of DNNs will converge to a linear SVM. Also, Beise et al. (2018) showed that the decision regions of DNNs with width smaller than the input dimension are unbounded.

More recently, and due to the popularity of the piecewise linear ReLU as an activation function, there has been a surge in the number of works that study this class of DNNs in particular. As a result, this has incited significant interest in new mathematical tools that help analyze piecewise linear functions, such as tropical geometry. While tropical geometry has shown its potential in many applications such as dynamic programming (Joswig & Schröter, 2019), linear programming (Allamigeon et al., 2015), multi-objective discrete optimization (Joswig & Loho, 2019), enumerative geometry (Mikhalkin, 2004), and economics (Akian et al., 2009; Mai Tran & Yu, 2015), it has only been recently used

---

[1] Code regenerating all our experiments is attached in the supplementary material.

to analyze DNNs. For instance, the work of Zhang et al. (2018) showed an equivalency between the family of DNNs with piecewise linear activations and integer weight matrices and the family of tropical rational maps, *i.e.* ratio between two multi-variate polynomials in tropical algebra. This study was mostly concerned about characterizing the complexity of a DNN by counting the number of linear regions, into which the function represented by the DNN can divide the input space. This was done by counting the number of vertices of a polytope representation recovering the results of Montufar et al. (2014) with a simpler analysis. More recently, Smyrnis & Maragos (2019) leveraged this equivalency to propose a heuristic for neural network minimization through approximating the tropical rational map.

**Contributions.** In this paper, we take the results of Zhang et al. (2018) several steps further and present a novel perspective on the decision boundaries of DNNs using tropical geometry. To that end, our contributions are three-fold. **(i)** We derive a geometric representation (convex hull between two zonotopes) for a super set to the decision boundaries of a DNN in the form (Affine, ReLU, Affine). **(ii)** We demonstrate a support for the lottery ticket hypothesis (Frankle & Carbin, 2019) from a geometric perspective. **(iii)** We leverage the geometric representation of the decision boundaries, referred to as the decision boundaries polytope, in two interesting applications: network pruning and adversarial attacks. For *tropical pruning*, we design a geometrically inspired optimization to prune the parameters of a given network such that the decision boundaries polytope of the pruned network does not deviate too much from its original network counterpart. We conduct extensive experiments with AlexNet (Krizhevsky et al., 2012) and VGG16 (Simonyan & Zisserman, 2014) on SVHN (Netzer et al., 2011), CIFAR10, and CIFAR 100 (Krizhevsky & Hinton, 2009) datasets, in which 90% pruning rate is achieved with a marginal drop in testing accuracy. For *tropical adversarial attacks*, we show that one can construct input adversaries that can change network predictions by perturbing the decision boundaries polytope.

## 2 PRELIMINARIES TO TROPICAL GEOMETRY

For completeness, we first provide preliminaries to tropical geometry (Itenberg et al., 2009; Maclagan & Sturmfels, 2015).

**Definition 1.** *(Tropical Semiring[2]) The tropical semiring $\mathbb{T}$ is the triplet $\{\mathbb{R} \cup \{-\infty\}, \oplus, \odot\}$, where $\oplus$ and $\odot$ define tropical addition and tropical multiplication, respectively. They are denoted as:*

$$x \oplus y = \max\{x, y\}, \qquad x \odot y = x + y, \qquad \forall x, y \in \mathbb{T}.$$

*It can be readily shown that $-\infty$ is the additive identity and $0$ is the multiplicative identity.*

Given the previous definition, a tropical power can be formulated as $x^{\odot a} = x \odot x \cdots \odot x = a.x$, for $x \in \mathbb{T}$, $a \in \mathbb{N}$, where $a.x$ is standard multiplication. Moreover, a tropical quotient can be defined as: $x \oslash y = x - y$, where $x - y$ is standard subtraction. For ease of notation, we write $x^{\odot a}$ as $x^a$.

**Definition 2.** *(Tropical Polynomials) For $\mathbf{x} \in \mathbb{T}^d$, $c_i \in \mathbb{R}$ and $\mathbf{a}_i \in \mathbb{N}^d$, a d-variable tropical polynomial with $n$ monomials $f : \mathbb{T}^d \to \mathbb{T}^d$ can be expressed as:*

$$f(\mathbf{x}) = (c_1 \odot \mathbf{x}^{\mathbf{a}_1}) \oplus (c_2 \odot \mathbf{x}^{\mathbf{a}_2}) \oplus \cdots \oplus (c_n \odot \mathbf{x}^{\mathbf{a}_n}), \qquad \forall \ \mathbf{a}_i \neq \mathbf{a}_j \ \text{when} \ i \neq j.$$

*We use the more compact vector notation $\mathbf{x}^{\mathbf{a}} = x_1^{a_1} \odot x_2^{a_2} \cdots \odot x_d^{a_d}$. Moreover and for ease of notation, we will denote $c_i \odot \mathbf{x}^{\mathbf{a}_i}$ as $c_i \mathbf{x}^{\mathbf{a}_i}$ throughout the paper.*

**Definition 3.** *(Tropical Rational Functions) A tropical rational is a standard difference or a tropical quotient of two tropical polynomials: $f(\mathbf{x}) - g(\mathbf{x}) = f(\mathbf{x}) \oslash g(\mathbf{x})$.*

Algebraic curves or hypersurfaces in algebraic geometry, which are the solution sets to polynomials, can be analogously extended to tropical polynomials too.

**Definition 4.** *(Tropical Hypersurfaces) A tropical hypersurface of a tropical polynomial $f(\mathbf{x}) = c_1 \mathbf{x}^{\mathbf{a}_1} \oplus \cdots \oplus c_n \mathbf{x}^{\mathbf{a}_n}$ is the set of points $\mathbf{x}$ where $f$ is attained by two or more monomials in $f$, i.e.*

$$\mathcal{T}(f) := \{\mathbf{x} \in \mathbb{R}^d : c_i \mathbf{x}^{\mathbf{a}_i} = c_j \mathbf{x}^{\mathbf{a}_j} = f(\mathbf{x}), \ \text{for some} \ \mathbf{a}_i \neq \mathbf{a}_j\}.$$

Tropical hypersurfaces divide the domain of $f$ into convex regions, where $f$ is linear in each region. Also, every tropical polynomial can be associated with a Newton polytope.

---

[2]A semiring is a ring that lacks an additive inverse.

Figure 1: **Tropical Hypersurfaces and their Corresponding Dual Subdivisions.** We show three tropical polynomials, where the solid red and black lines are the tropical hypersurfaces $\mathcal{T}(f)$ and dual subdivisions $\delta(f)$ to the corresponding tropical polynomials, respectively. $\mathcal{T}(f)$ divides The domain of $f$ into convex regions where $f$ is linear. Moreover, each region is in one-to-one correspondence with each node of $\delta(f)$. Lastly, the tropical hypersurfaces are parallel to the normals of the edges of $\delta(f)$ shown by dashed red lines.

**Definition 5.** *(Newton Polytopes) The Newton polytope of a tropical polynomial $f(\mathbf{x}) = c_1 \mathbf{x}^{\mathbf{a}_1} \oplus \cdots \oplus c_n \mathbf{x}^{\mathbf{a}_n}$ is the convex hull of the exponents $\mathbf{a}_i \in \mathbb{N}^d$ regarded as points in $\mathbb{R}^d$, i.e.*

$$\Delta(f) := ConvHull\{\mathbf{a}_i \in \mathbb{R}^d : i = 1, \dots, n \text{ and } c_i \neq -\infty\}.$$

A tropical polynomial determines a dual subdivision, which can be constructed by projecting the collection of upper faces (UF) in $\mathcal{P}(f) := \text{ConvHull}\{(\mathbf{a}_i, c_i) \in \mathbb{R}^d \times \mathbb{R} : i = 1, \dots, n\}$ onto $\mathbb{R}^d$. That is to say, the dual subdivision determined by $f$ is given as $\delta(f) := \{\pi(p) \subset \mathbb{R}^d : p \in \text{UF}(\mathcal{P}(f))\}$, where $\pi : \mathbb{R}^d \times \mathbb{R} \to \mathbb{R}^d$ is the projection that drops the last coordinate. It has been shown by Maclagan & Sturmfels (2015) that the tropical hypersurface $\mathcal{T}(f)$ is the $(d\text{-}1)$-skeleton of the polyhedral complex dual to $\delta(f)$. This implies that each node of the dual subdivision $\delta(f)$ corresponds to one region in $\mathbb{R}^d$ where $f$ is linear. This is exemplified in Figure 1 with three tropical polynomials, and to see this clearly, we will elaborate on the first tropical polynomial example $f(x, y) = x \oplus y \oplus 0$. Note that as per Definition 4, the tropical hypersurface is the set of points $(x, y)$ where $x = y, y = 0$, and $x = 0$. This indeed gives rise to the three solid red lines indicating the tropical hypersurfaces. As for the dual subdivision $\delta(f)$, we observe that $x \oplus y \oplus 0$ can be written as $(x^1 \odot y^0) \oplus (x^0 \odot y^1) \oplus (x^0 \odot y^0)$. Thus, and since the monomials are bias free ($c_i = 0$), then $\mathcal{P}(f) = \text{ConvHull}\{(1, 0, 0), (0, 1, 0), (0, 0, 0)\}$. It is then easy to see that $\delta(f) = \text{ConvHull}\{(1, 0), (0, 1), (0, 0)\}$, since $\text{UP}(\mathcal{P}(f)) = \mathcal{P}(f)$, which is the black triangle in solid lines in Figure 1. One key observation in all three examples in Figure 1 is that the number of regions where $f$ is linear (that is 3, 6 and 10, respectively) is equal to the number of nodes in the corresponding dual subdivisions. Second, the tropical hypersurfaces are parallel to the normals to the edges of the dual subdivision polytope. This observation will be essential for the remaining part of the paper. Several other observations are summarized by Brugallé & Shaw (2014). Moreover, Zhang et al. (2018) showed an equivalency between tropical rational maps and a family of neural network $f : \mathbb{R}^n \to \mathbb{R}^k$ with piecewise linear activations through the following theorem.

**Theorem 1.** *(Tropical Characterization of Neural Networks, (Zhang et al., 2018)). A feedforward neural network with integer weights and real biases with piecewise linear activation functions is a function $f : \mathbb{R}^n \to \mathbb{R}^k$, whose coordinates are tropical rational functions of the input, i.e., $f(\mathbf{x}) = H(\mathbf{x}) \oslash Q(\mathbf{x}) = H(\mathbf{x}) - Q(\mathbf{x})$, where $H$ and $Q$ are tropical polynomials.*

While this is new in the context of tropical geometry, it is not surprising, since any piecewise linear function can be written as a difference of two max functions over a set of hyperplanes (Melzer, 1986).

Before any further discussion, we first recap the definition of zonotopes.

**Definition 6.** *Let $\mathbf{u}^1, \dots, \mathbf{u}^L \in \mathbb{R}^n$. The zonotope formed by $\mathbf{u}^1, \dots, \mathbf{u}^L$ is defined as $\mathcal{Z}(\mathbf{u}^1, \dots, \mathbf{u}^L) := \{\sum_{i=1}^L x_i \mathbf{u}^i : 0 \leq x_i \leq 1\}$. Equivalently, $\mathcal{Z}$ can be expressed with respect to the generator matrix $\mathbf{U} \in \mathbb{R}^{L \times n}$, where $\mathbf{U}(i, :) = {\mathbf{u}^i}^\top$ as $\mathcal{Z}_{\mathbf{U}} := \{\mathbf{U}^\top \mathbf{x} : \forall \mathbf{x} \in [0, 1]^L\}$.*

Another common definition for a zonotope is the Minkowski sum of the set of line segments $\{\mathbf{u}^1, \dots, \mathbf{u}^L\}$ (refer to **appendix**), where a line segment of the vector $\mathbf{u}^i$ in $\mathbb{R}^n$ is defined as $\{\alpha \mathbf{u}^i : \forall \alpha \in [0, 1]\}$. It is well-known that the number of vertices of a zonotope is polynomial in the number of line segments, *i.e.* $|\text{vert}(\mathcal{Z}_{\mathbf{U}})| \leq 2 \sum_{i=0}^{n-1} \binom{L-1}{i} = \mathcal{O}(L^{n-1})$ (Gritzmann & Sturmfels, 1993).

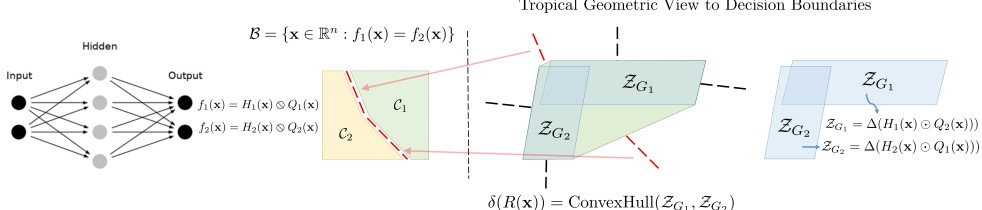

Figure 2: **Decision Boundaries as Geometric Structures.** The decision boundaries $\mathcal{B}$ (in red) comprise two linear pieces separating classes $\mathcal{C}_1$ and $\mathcal{C}_2$. As per Theorem 2, the dual subdivision of this single hidden neural network is the convex hull between the zonotopes $\mathcal{Z}_{\mathbf{G}_1}$ and $\mathcal{Z}_{\mathbf{G}_2}$. The normals to the dual subdivison $\delta(R(\mathbf{x}))$ are in one-to-one correspondence to the tropical hypersurface $\mathcal{T}(R(\mathbf{x}))$, which is a superset to the decision boundaries $\mathcal{B}$. Note that some of the normals to $\delta(R(\mathbf{x}))$ (in red) are parallel to the decision boundaries.

## 3    DECISION BOUNDARIES OF NEURAL NETWORKS AS POLYTOPES[3]

In this section, we analyze the decision boundaries of a network in the form (Affine, ReLU, Affine) using tropical geometry. For ease, we use ReLUs as the non-linear activation, but any other piecewise linear function can also be used. The functional form of this network is: $f(\mathbf{x}) = \mathbf{B}\max(\mathbf{A}\mathbf{x} + \mathbf{c}_1, \mathbf{0}) + \mathbf{c}_2$, where $\max(.)$ is an element-wise operator. The outputs of the network $f$ are the logit scores. Throughout this section, we assume[4] that $\mathbf{A} \in \mathbb{Z}^{p \times n}$, $\mathbf{B} \in \mathbb{Z}^{2 \times p}$, $\mathbf{c}_1 \in \mathbb{R}^p$ and $\mathbf{c}_2 \in \mathbb{R}^2$. For ease of notation, we only consider networks with two outputs, *i.e.* $\mathbf{B}^{2 \times p}$, where the extension to a multi-class output follows naturally and is discussed in the **appendix**. Now, since $f$ is a piecewise linear function, each output can be expressed as a tropical rational as per Theorem 1. If $f_1$ and $f_2$ refer to the first and second outputs respectively, we have $f_1(\mathbf{x}) = H_1(\mathbf{x}) \oslash Q_1(\mathbf{x})$ and $f_2(\mathbf{x}) = H_2(\mathbf{x}) \oslash Q_2(\mathbf{x})$, where $H_1, H_2, Q_1$ and $Q_2$ are tropical polynomials. In what follows and for ease of presentation, we present our main results where the network $f$ has no biases, *i.e.* $\mathbf{c}_1 = \mathbf{0}$ and $\mathbf{c}_2 = \mathbf{0}$, and we leave the generalization to the **appendix**.

**Theorem 2.** *For a bias-free neural network in the form* $f(\mathbf{x}) : \mathbb{R}^n \to \mathbb{R}^2$*, where* $\mathbf{A} \in \mathbb{Z}^{p \times n}$ *and* $\mathbf{B} \in \mathbb{Z}^{2 \times p}$*, let* $R(\mathbf{x}) = H_1(\mathbf{x}) \odot Q_2(\mathbf{x}) \oplus H_2(\mathbf{x}) \odot Q_1(\mathbf{x})$ *be a tropical polynomial. Then:*

- *Let* $\mathcal{B} = \{\mathbf{x} \in \mathbb{R}^n : f_1(\mathbf{x}) = f_2(\mathbf{x})\}$ *define the decision boundaries of* $f$*, then* $\mathcal{B} \subseteq \mathcal{T}(R(\mathbf{x}))$.

- $\delta(R(\mathbf{x})) = ConvHull(\mathcal{Z}_{\mathbf{G}_1}, \mathcal{Z}_{\mathbf{G}_2})$. $\mathcal{Z}_{\mathbf{G}_1}$ *is a zonotope in* $\mathbb{R}^n$ *with line segments* $\{(\mathbf{B}^+(1,j) + \mathbf{B}^-(2,j))[\mathbf{A}^+(j,:), \mathbf{A}^-(j,:)]\}_{j=1}^p$ *and shift* $(\mathbf{B}^-(1,:) + \mathbf{B}^+(2,:))\mathbf{A}^-$*, where* $\mathbf{A}^+ = \max(\mathbf{A}, 0)$ *and* $\mathbf{A}^- = \max(-\mathbf{A}, 0)$*.* $\mathcal{Z}_{\mathbf{G}_2}$ *is a zonotope in* $\mathbb{R}^n$ *with line segments* $\{(\mathbf{B}^-(1,j) + \mathbf{B}^+(2,j))[\mathbf{A}^+(j,:), \mathbf{A}^-(j,:)]\}_{j=1}^p$ *and shift* $(\mathbf{B}^+(1,:) + \mathbf{B}^-(2,:))\mathbf{A}^-$.

**Digesting Theorem 2.** This theorem aims at characterizing the decision boundaries (where $f_1(\mathbf{x}) = f_2(\mathbf{x})$) of a bias-free neural network of the form (Affine, ReLU, Affine) through the lens of tropical geometry. In particular, the first result of Theorem 2 states that the tropical hypersurface $\mathcal{T}(R(\mathbf{x}))$ of the tropical polynomial $R(\mathbf{x})$ is a superset to the set of points forming the decision boundaries, *i.e.* $\mathcal{B}$. Just as discussed earlier and exemplified in Figure 1, tropical hypersurfaces are associated with a corresponding dual subdivision polytope $\delta(R(\mathbf{x}))$. Based on this, the second result of Theorem 2 states that this dual subdivision is precisely the convex hull of two zonotopes denoted as $\mathcal{Z}_{\mathbf{G}_1}$ and $\mathcal{Z}_{\mathbf{G}_2}$, where each zonotope is only a function of the network parameters $\mathbf{A}$ and $\mathbf{B}$.

Theorem 2 bridges the gap between the behaviour of the decision boundaries $\mathcal{B}$, through the superset $\mathcal{T}(R(\mathbf{x}))$, and the polytope $\delta(R(\mathbf{x}))$, which is the convex hull of two zonotopes. It is worthwhile to mention that Zhang et al. (2018) discussed a special case of the first part of Theorem 2 for a neural network with a single output and a score function $s(\mathbf{x})$ to classify the output. To the best of our knowledge, this work is the first to propose a tropical geometric formulation of a superset containing the decision boundaries of a multi-class classification neural network. In particular, the first result of Theorem 2 states that one can perhaps study the decision boundaries, $\mathcal{B}$, directly by studying their superset $\mathcal{T}(R(\mathbf{x}))$. While studying $\mathcal{T}(R(\mathbf{x}))$ can be equally difficult, the second result of Theorem 2 comes in handy. First, note that, since the network is bias-free, $\pi$ becomes an identity mapping with $\delta(R(\mathbf{x})) = \Delta(R(\mathbf{x}))$, and thus the dual subdivision $\delta(R(\mathbf{x}))$, which is the Newton polytope $\Delta(R(\mathbf{x}))$ in this case, becomes a well-structured geometric object that can be exploited to preserve

---

[3]All proofs are left for the **appendix**.

[4]Without loss of generality, as one can very well approximate real weights as fractions and multiply by the least common multiple of the denominators as discussed in Zhang et al. (2018).

Figure 3: **Effect of Different Initializations on the Decision Boundaries Polytope.** From left to right: training dataset, decision boundaries polytope of original network (before pruning), followed by the decision boundaries polytope for networks pruned at different pruning percentages using different initializations. Note that in the original polytope there are many more vertices than just 4, but they are very close to each other forming many small edges that are not visible in the figure.

decision boundaries as per the second part of Theorem 2. Now, based on the results of Maclagan & Sturmfels (2015) (Proposition 3.1.6) and as discussed in Figure 1, the normals to the edges of the polytope $\delta(R(\mathbf{x}))$ (convex hull of two zonotopes) are in one-to-one correspondence with the tropical hypersurface $\mathcal{T}(R(\mathbf{x}))$. Therefore, one can study the decision boundaries, or at least their superset $\mathcal{T}(R(\mathbf{x}))$, by studying the orientation of the dual subdivision $\delta(R(\mathbf{x}))$.

While Theorem 2 presents a strong relation between a polytope (convex hull of two zonotopes) and the decision boundaries, it remains unclear how such a polytope can be efficiently constructed. Although the number of vertices of a zonotope is polynomial in the number of its generating line segments, fast algorithms for enumerating these vertices are still restricted to zonotopes with line segments starting at the origin Stinson et al. (2016). Since the line segments generating the zonotopes in Theorem 2 have arbitrary end points, we present the next result that transforms these line segments into a generator matrix of line segments starting from the origin as in Definition 6. This result is essential for an efficient computation of the zonotopes in Theorem 2.

**Proposition 1.** *The zonotope formed by $p$ line segments in $\mathbb{R}^n$ with arbitrary end points $\{[\mathbf{u}_1^i, \mathbf{u}_2^i]\}_{i=1}^p$ is equivalent to the zonotope formed by the line segments $\{[\mathbf{u}_1^i - \mathbf{u}_2^i, \mathbf{0}]\}_{i=1}^p$ with a shift of $\sum_{i=1}^p \mathbf{u}_2^i$.*

We can now represent with the following corollary the arbitrary end point line segments forming the zonotopes in Theorem 2 with generator matrices, which allow us to leverage existing algorithms that enumerate zonotope vertices Stinson et al. (2016).

**Corollary 1.** *The generators of $\mathcal{Z}_{\mathbf{G}_1}, \mathcal{Z}_{\mathbf{G}_2}$ in Theorem 2 can be defined as $\mathbf{G}_1 = Diag[(\mathbf{B}^+(1,:)) + (\mathbf{B}^-(2,:))]\mathbf{A}$ and $\mathbf{G}_2 = Diag[(\mathbf{B}^+(2,:)) + (\mathbf{B}^-(1,:))]\mathbf{A}$, both with shift $(\mathbf{B}^-(1,:) + \mathbf{B}^+(2,:) + \mathbf{B}^+(1,:) + \mathbf{B}^-(2,:)) \mathbf{A}^-$, where $Diag(\mathbf{v})$ arranges $\mathbf{v}$ in a diagonal matrix.*

Next, we show several applications for Theorem 2 by leveraging the tropical geometric structure.

## 4 TROPICAL PERSPECTIVE TO THE LOTTERY TICKET HYPOTHESIS

The lottery ticket hypothesis was recently proposed by Frankle & Carbin (2019), in which the authors surmise the existence of sparse trainable sub-networks of dense, randomly-initialized, feed-forward networks that when trained in isolation perform as well as the original network in a similar number of iterations. To find such sub-networks, Frankle & Carbin (2019) propose the following simple algorithm: perform standard network pruning, initialize the pruned network with the same initialization that was used in the original training setting, and train with the same number of epochs. They hypothesize that this results in a smaller network with a similar accuracy. In other words, a sub-network can have decision boundaries similar to those of the original larger network. While we do not provide a theoretical reason why this pruning algorithm performs favorably, we utilize the geometric structure that arises from Theorem 2 to reaffirm such behaviour. In particular, we show that the orientation of the dual subdivision $\delta(R(\mathbf{x}))$ (referred to as decision boundaries polytope), where the normals to its edges are parallel to $\mathcal{T}(R(\mathbf{x}))$ that is a superset to the decision boundaries, is preserved after pruning with the proposed initialization algorithm of Frankle & Carbin (2019). Conversely, pruning with a different initialization at each iteration results in a significant variation in the orientation of the decision boundaries polytope and ultimately in reduced accuracy.

To this end, we train a neural network with 2 inputs ($n = 2$), 2 outputs, and a single hidden layer with 40 nodes ($p = 40$). We then prune the network by removing the smallest $x\%$ of the weights. The pruned network is then trained using different initializations: (i) the same initialization as the original network (Frankle & Carbin, 2019), (ii) Xavier (Glorot & Bengio, 2010), (iii) standard Gaussian, and

(iv) zero mean Gaussian with variance 0.1. Figure 3 shows the decision boundaries polytope, *i.e.* $\delta(R(\mathbf{x}))$, as we perform more pruning (increasing the $x\%$) with different initializations. First, we show the decision boundaries by sampling and classifying points in a grid with the trained network (first subfigure). We then plot the decision boundaries polytope $\delta(R(\mathbf{x}))$ as per the second part of Theorem 2 denoted as original polytope (second subfigure). While there are many overlapping vertices in the original polytope, the normals to some of the edges (the major visible edges) are parallel to the decision boundaries shown in the first subfigure of Figure 3. We later show the decision boundaries polytope for the same network under different levels of pruning. One can observe that the orientation of $\delta(R(\mathbf{x}))$ for all different initialization schemes deviates much more from the original polytope as compared to the lottery ticket initialization. This gives an indication that lottery ticket initialization indeed preserves the decision boundaries, since it preserves the orientation of the decision boundaries polytope throughout the evolution of pruning. An alternative means to study the lottery ticket could be to directly observe the polytopes representing the functional form of the network, *i.e.* $\delta(H_{\{1,2\}}(\mathbf{x}))$ and $\delta(Q_{\{1,2\}}(\mathbf{x}))$, in lieu of the decision boundaries polytopes. However, this strategy may fail to provide a conclusive analysis of the lottery ticket, since there can exist multiple polytopes $\delta(H_{\{1,2\}}(\mathbf{x}))$ and $\delta(Q_{\{1,2\}}(\mathbf{x}))$ for networks with the same decision boundaries. This highlights the importance of studying the decision boundaries directly. Additional discussions and experiments are left for the **appendix**.

## 5 TROPICAL NETWORK PRUNING

Network pruning has been identified as an effective approach to reduce the computational cost and memory usage during network inference. While it dates back to the work of LeCun et al. (1990) and Hassibi & Stork (1993), network pruning has recently gained more attention. This is due to the fact that most neural networks over-parameterize commonly used datasets. In network pruning, the task is to find a smaller subset of the network parameters, such that the resulting smaller network has similar decision boundaries (and thus supposedly similar accuracy) to the original over-parameterized network. In this section, we show a new geometric approach towards network pruning. In particular and as indicated by Theorem 2, preserving the polytope $\delta(R(\mathbf{x}))$ preserves a superset to the decision boundaries, $\mathcal{T}(R(\mathbf{x}))$, and thus the decision boundaries themselves.

**Motivational Insight.** For a single hidden layer neural network, the dual subdivision to the decision boundaries is the polytope that is the convex hull of two zonotopes, where each is formed by taking the Minkowski sum of line segments (Theorem 2). Figure 4 shows an example, where pruning a neuron in the network has no effect on the dual subdivision polytope and hence no effect on performance. This occurs, since the tropical hypersurface $\mathcal{T}(R(\mathbf{x}))$ before and after pruning is preserved, thus, keeping the decision boundaries the same.

**Problem Formulation.** In light of the motivational insight, a natural question arises: *Given an over-parameterized binary output neural network $f(\mathbf{x}) = \mathbf{B}\max(\mathbf{Ax}, \mathbf{0})$, can one construct a new neural network, parameterized by sparser weight matrices $\tilde{\mathbf{A}}$ and $\tilde{\mathbf{B}}$, such that this smaller network has a dual subdivision $\delta(\tilde{R}(\mathbf{x}))$ that preserves the decision boundaries of the original network?*

To address this question, we propose the following optimization problem to compute $\tilde{\mathbf{A}}$ and $\tilde{\mathbf{B}}$:

$$\min_{\tilde{\mathbf{A}},\tilde{\mathbf{B}}} d\Big(\delta(\tilde{R}(\mathbf{x})),\delta(R(\mathbf{x}))\Big) = \min_{\tilde{\mathbf{A}},\tilde{\mathbf{B}}} d\Big(\text{ConvHull}\left(\mathcal{Z}_{\tilde{\mathbf{G}}_1},\mathcal{Z}_{\tilde{\mathbf{G}}_2}\right),\text{ConvHull}\left(\mathcal{Z}_{\mathbf{G}_1},\mathcal{Z}_{\mathbf{G}_2}\right)\Big). \quad (1)$$

The function $d(.)$ defines a distance between two geometric objects. Since the generators $\tilde{\mathbf{G}}_1$ and $\tilde{\mathbf{G}}_2$ are functions of $\tilde{\mathbf{A}}$ and $\tilde{\mathbf{B}}$ (as per Theorem 2), this optimization problem can be challenging to solve. However, for pruning purposes, one can observe from Theorem 2 that if the generators $\tilde{\mathbf{G}}_1$ and $\tilde{\mathbf{G}}_2$ had fewer number of line segments (rows), this corresponds to a fewer number of rows in the weight matrix $\tilde{\mathbf{A}}$ (sparser weights). So, we observe that if $\tilde{\mathbf{G}}_1 \approx \mathbf{G}_1$ and $\tilde{\mathbf{G}}_2 \approx \mathbf{G}_2$, then $\delta(\tilde{R}(\mathbf{x})) \approx \delta(R(\mathbf{x}))$, and thus the decision boundaries tend to be preserved as a consequence. Therefore, we propose the following optimization problem as a surrogate to the one in Problem (1):

$$\min_{\tilde{\mathbf{A}},\tilde{\mathbf{B}}} \frac{1}{2}\Big(\left\|\tilde{\mathbf{G}}_1 - \mathbf{G}_1\right\|_F^2 + \left\|\tilde{\mathbf{G}}_2 - \mathbf{G}_2\right\|_F^2\Big) + \lambda_1\left\|\tilde{\mathbf{G}}_1\right\|_{2,1} + \lambda_2\left\|\tilde{\mathbf{G}}_2\right\|_{2,1}. \quad (2)$$

The matrix mixed norm for $\mathbf{C} \in \mathbb{R}^{n\times k}$ is defined as $\|\mathbf{C}\|_{2,1} = \sum_{i=1}^{n}\|\mathbf{C}(i,:)\|_2$, which encourages the matrix $\mathbf{C}$ to be row sparse, *i.e.* complete rows of $\mathbf{C}$ are zero. The first two terms in Problem

$$\delta(R(\mathbf{x})) = \text{ConvexHull}(\mathcal{Z}_{G_1}, \mathcal{Z}_{G_2}) \quad \delta(\bar{R}(\mathbf{x})) = \text{ConvexHull}(\mathcal{Z}_{\bar{G}_1}, \mathcal{Z}_{\bar{G}_2})$$

Figure 4: **Tropical Pruning Pipeline.** Pruning the $4^{th}$ node, or equivalently removing the two yellow vertices of zonotope $\mathcal{Z}_{G_2}$ does not affect the decision boundaries polytope, which will lead to no change in accuracy.

(2) aim at approximating the original dual subdivision $\delta(R(\mathbf{x}))$ by approximating the underlying generator matrices, $\mathbf{G}_1$ and $\mathbf{G}_2$. This aims to preserve the orientation of the decision boundaries of the newly constructed network. On the other hand, the second two terms in Problem (2) act as regularizers to control the sparsity of the constructed network by controlling the sparsity in the number of line segments. We observe that Problem (2) is not quadratic in its variables, since as per Corollary, 1 $\tilde{\mathbf{G}}_1 = \text{Diag}[\text{ReLU}(\tilde{\mathbf{B}}(1,:)) + \text{ReLU}(-\tilde{\mathbf{B}}(2,:))]\tilde{\mathbf{A}}$ and $\tilde{\mathbf{G}}_2 = \text{Diag}[\text{ReLU}(\tilde{\mathbf{B}}(2,:)) + \text{ReLU}(-\tilde{\mathbf{B}}(1,:))]\tilde{\mathbf{A}}$. However, since Problem (2) is separable in the rows of $\tilde{\mathbf{A}}$ and $\tilde{\mathbf{B}}$, we solve Problem (2) via alternating optimization over these rows, where each sub-problem can be shown to be convex and exhibits a closed-form solution leading to a very efficient solver. For ease of notation, we refer to $\text{ReLU}(\tilde{\mathbf{B}}(i,:))$ and $\text{ReLU}(-\tilde{\mathbf{B}}(i,:))$ as $\tilde{\mathbf{B}}^+(i,:)$ and $\tilde{\mathbf{B}}^-(i,:)$, respectively. As such, the per row update for $\tilde{\mathbf{A}}$ (first linear layer) is given as follows:

$$\tilde{\mathbf{A}}(i,:) = \max\left(1 - \frac{1}{2}\frac{\lambda_1\sqrt{\mathbf{c}_1^i} + \lambda_2\sqrt{\mathbf{c}_2^i}}{\frac{1}{2}(\mathbf{c}_1^i + \mathbf{c}_2^i)}\frac{1}{\left\|\frac{\mathbf{c}_1^i\mathbf{G}_1(i,:) + \mathbf{c}_2^i\mathbf{G}_2(i,:)}{\frac{1}{2}(\mathbf{c}_1^i+\mathbf{c}_2^i)}\right\|_2}, 0\right)\left(\frac{\mathbf{c}_1^i\mathbf{G}_1(i,:) + \mathbf{c}_2^i\mathbf{G}_2(i,:)}{\frac{1}{2}(\mathbf{c}_1^i + \mathbf{c}_2^i)}\right),$$

where $\mathbf{c}_1^i$ is the $i^{th}$ element of $\mathbf{c}_1 = \text{ReLU}(\mathbf{B}(1,:)) + \text{ReLU}(-\mathbf{B}(2,:))$ and $\mathbf{c}_2 = \text{ReLU}(\mathbf{B}(2,:)) + \text{ReLU}(-\mathbf{B}(1,:))$. Similarly, the closed form update for the $j^{th}$ element of the second linear layer is as follows:

$$\tilde{\mathbf{B}}^+(1,j) = \max\left(0, \frac{\tilde{\mathbf{A}}(j,:)^\top\tilde{\mathbf{G}}_{1+}(j,:) - \lambda\|\tilde{\mathbf{A}}(j,:)\|_2}{\|\tilde{\mathbf{A}}(j,:)\|_2^2}\right),$$

where $\mathbf{G}_{1+} = \text{Diag}(\mathbf{B}^+(1,:))\mathbf{A}$. A similar argument can be used to update the variables $\tilde{\mathbf{B}}^+(2,:)$, $\tilde{\mathbf{B}}^-(1,:)$, and $\tilde{\mathbf{B}}^-(2,:)$. The details of deriving the aforementioned update steps and the extension to the multi-class case are left to the **appendix**. Note that all updates are cheap, as they are expressed in a closed form single step. In all subsequent experiments, we find that running the alternating optimization for a single iteration is sufficient to converge to a reasonable solution, thus, leading to a very efficient overall solver.

*Extension to Deeper Networks.* While the theoretical results in Theorem 2 and Corollary 1 only hold for a shallow network in the form of (Affine, ReLU, Affine), we propose a greedy heuristic to prune much deeper networks by applying the aforementioned optimization for consecutive blocks of (Affine, ReLU, Affine) starting from the input and ending at the output of the network. This extension from a theoretical study of 2 layer network was observed in several works such as (Bibi et al., 2018).

**Experiments on Tropical Pruning.** Here, we evaluate the performance of the proposed pruning approach as compared to several classical approaches on several architectures and datasets. In particular, we compare our tropical pruning approach against Class Blind (CB), Class Uniform (CU) and Class Distribution (CD) (Han et al., 2015; See et al., 2016). In Class Blind, all the parameters of a layer are sorted by magnitude where the $x\%$ with smallest magnitude are pruned. In contrast, Class Uniform prunes the parameters with smallest $x\%$ magnitudes per node in a layer. Lastly, Class Distribution performs pruning of all parameters for each node in the layer, just as in Class Uniform, but the parameters are pruned based on the standard deviation $\sigma_c$ of the magnitude of the parameters per node. Since fully connected layers in deep neural networks tend to have much higher memory complexity than convolutional layers, we restrict our focus to pruning fully connected layers. We train AlexNet and VGG16 on SVHN, CIFAR10, and CIFAR100 datasets. We observe that we can prune more than 90% of the classifier parameters for both networks without affecting the accuracy. Since pruning is often a single block within a larger compression scheme that in many cases involves inexpensive fast fine tuning, we demonstrate experimentally that our approach can is competitive

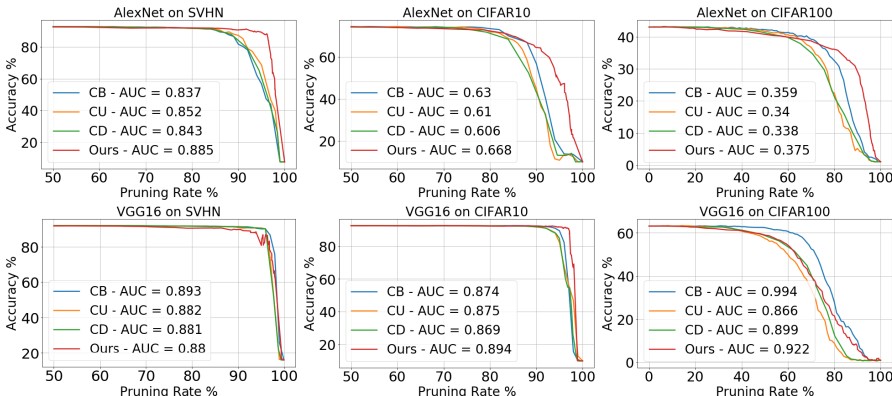

Figure 5: **Results of Tropical Pruning.** Pruning-accuracy plots for AlexNet (top) and VGG16 (bottom) trained on SVHN, CIFAR10, and CIFAR100, pruned with our tropical method and three other pruning methods.

and sometimes outperforms other methods even when all parameters or when only the biases are fine-tuned after pruning. These experiments in addition to many others are left for the **appendix**.

*Setup.* To account for the discrepancy in input resolution, we adapt the architectures of AlexNet and VGG16, since they were originally trained on ImageNet (Deng et al., 2009). The fully connected layers of AlexNet and VGG16 have sizes (256,512,10) and (512,512,10) for SVHN and CIFAR10, respectively, and with the last dimension increased to 100 for CIFAR100. All networks were trained to baseline test accuracy of (92%,74%,43%) for AlexNet on SVHN, CIFAR10, and CIFAR100, respectively and (92%,92%,70%) for VGG16. To evaluate the performance of pruning and following previous work Han et al. (2015), we report the area under the curve (AUC) of the pruning-accuracy plot. The higher the AUC is, the better the trade-off is between pruning rate and accuracy. For efficiency purposes, we run the optimization in Problem 2 for a single alternating iteration to identify the rows in $\tilde{\mathbf{A}}$ and elements of $\tilde{\mathbf{B}}$ that will be pruned.

*Results.* Figure 5 shows the comparison between our tropical approach and the three popular pruning schemes on both AlexNet and VGG16 over the different datasets. Our proposed approach can indeed prune out as much as 90% of the parameters of the classifier without sacrificing much of the accuracy. For AlexNet, we achieve much better performance in pruning as compared to other methods. In particular, we are better in AUC by 3%, 3%, and 2% over other pruning methods on SVHN, CIFAR10 and CIFAR100, respectively. This indicates that the decision boundaries can indeed be preserved by preserving the dual subdivision polytope. For VGG16, we perform similarly well on both SVHN and CIFAR10 and slightly worse on CIFAR100. While the performance achieved here is comparable to the other pruning schemes, if not better, we emphasize that our contribution does not lie in outperforming state-of-the-art pruning methods, but in giving a new geometry-based perspective to network pruning. More experiments were conducted where only network biases or only the classifier are fine-tuned after pruning. Retraining only biases can be sufficient, as they do not contribute to the orientation of the decision boundaries polytope (and effectively the decision boundaries), but only to its translation. Discussions on biases and more results are left for the **appendix**.

*Comparison Against Tropical Geometry Approaches.* A recent tropical geometry inspired approach was proposed to address the problem of network pruning. In particular, Smyrnis & Maragos (2019; 2020) (SM) proposed an interesting yet heuristic algorithm to directly approximate the tropical rational by approximating the Newton polytope. For fair comparison and following the setup of SM, we train LeNet on MNIST and monitor the test accuracy as we prune its neurons. We report (neurons kept, SM, ours) triplets in (%) as follows: (100, 98.60, 98.84), (90, 95.71, 98.82), (75, 95.05, 98.8), (50, 95.52, 98.71), (25, 91.04, 98.36), (10, 92.79, 97.99), and (5, 92.93, 94.91). It is clear that tropical pruning outperforms SM by a margin that reaches 7%. This demonstrates that our theoretically motivated approach is still superior to more recent pruning approaches.

## 6 TROPICAL ADVERSARIAL ATTACKS

DNNs are notorious for being sensitive to imperceptible noise at their inputs referred to as adversarial attacks. Several works investigated DNNs' decision boundaries in the presence of such adversaries. For instance, Khoury & Hadfield-Menell (2018) analyzed the high dimensional geometry of adversar-

ial examples by means of manifold reconstruction while He et al. (2018) crafted adversarial attacks by estimating the distance to the decision boundaries using random search directions. In this work, we show how Theorem 2 can be leveraged to construct a tropical geometric adversarial attack. Due to the space limitation, we leave the extensive formulation, the algorithm to find the adversary, and the experimental results on synthetic and real datasets to the **appendix**.

## 7 CONCLUSION

We leverage tropical geometry to characterize the decision boundaries of neural networks in the form (Affine, ReLU, Affine) and relate it to geometric objects such as zonotopes. We then provide a tropical perspective to support the lottery ticket hypothesis, prune networks, and design adversarial attacks. A natural extension is a compact derivation for the characterization of the decision boundaries of convolutional neural networks and graphical convolutional networks.

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

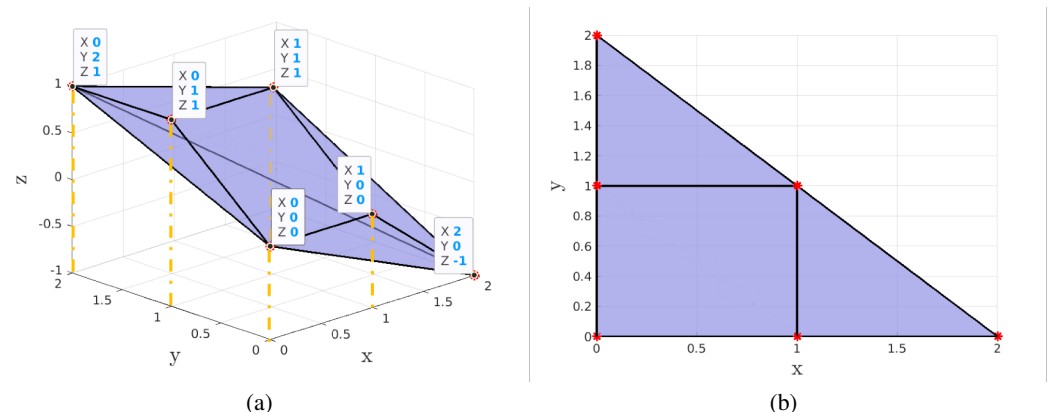

(a)                                      (b)

Figure 6: **The Newton Polygon and the Corresponding Dualsubdivision**. The Figure on the left shows the newton polygon $\mathcal{P}(f)$ for the tropical polynomial defined in the second example in Figure 1. The dual subdivision $\delta(f)$ is constructed by projecting the upper faces of $\mathcal{P}(f)$, shadowing, on $\mathbb{R}^2$.

## 8   PRELIMINARIES AND DEFINITIONS.

**Fact 1.** $P\tilde{+}Q = \{p + q, \forall p \in P$ and $q \in Q\}$ is the Minkowski sum between two sets $P$ and $Q$.

**Fact 2.** *Let $f$ be a tropical polynomial and let $a \in \mathbb{N}$. Then*

$$\mathcal{P}(f^a) = a\mathcal{P}(f).$$

Let both $f$ and $g$ be tropical polynomials. Then

**Fact 3.**
$$\mathcal{P}(f \odot g) = \mathcal{P}(f)\tilde{+}\mathcal{P}(g). \tag{3}$$

**Fact 4.**
$$\mathcal{P}(f \oplus g) = ConvexHull\Big(\mathcal{V}\left(\mathcal{P}(g)\right) \cup \mathcal{V}\left(\mathcal{P}(g)\right)\Big). \tag{4}$$

Note that $\mathcal{V}(\mathcal{P}(f))$ is the set of vertices of the polytope $\mathcal{P}(f)$.

**Definition 7.** *Upper Face of a Polytope $P$: $UF(P)$ is an upper face of polytope $P$ in $\mathbb{R}^n$ if $\mathbf{x}+t\mathbf{e}_n \notin P$ for any $\mathbf{x} \in UF(P)$, $t > 0$ where $\mathbf{e}_n$ is a canonical vector. Formally,*

$$UF(P) = \{\mathbf{x} : \ \mathbf{x} \in P, \ \mathbf{x} + t\mathbf{e}_n \notin P \quad \forall t > 0\}$$

## 9   EXAMPLES

We revise the second example in Figure 1. Note that the two dimensional tropical polynomial $f(x, y)$ can be written as follows:

$$
\begin{aligned}
f(x,y) &= (x \oplus y \oplus 0) \odot ((x \oslash 1) \oplus (y \odot 1) \oplus 0)\\
&= (x \odot x \oslash 1) \oplus (x \odot y \odot 1) \oplus (x \odot 0) \oplus (y \odot x \oslash 1) \oplus (y \odot y \odot 1)\\
&\quad \oplus (y \odot 0) \oplus (0 \odot x \oslash 1) \oplus (0 \odot y \odot 1) \oplus (0 \odot 0)\\
&= (x^2 \oslash 1) \oplus (x \odot y \odot 1) \oplus (x) \oplus (y \odot x \oslash 1) \oplus (y^2 \odot 1) \oplus (y) \oplus (x \oslash 1) \oplus (y \odot 1) \oplus (0)\\
&= (x^2 \oslash 1) \oplus (x \odot y \odot 1) \oplus (x) \oplus (y^2 \odot 1) \oplus (y \odot 1) \oplus (0)\\
&= (x^2 \odot y^0 \oslash 1) \oplus (x \odot y \odot 1) \oplus (x \odot y^0 \odot 0) \oplus (x^0 \odot y^2 \odot 1) \oplus (x^0 \odot y \odot 1) \oplus (x^0 \odot y^0 \odot 0)
\end{aligned}
$$

First equality follows since multiplication is distributive in rings and semi rings. The second equality follows since $0$ is the multiplication identity. The penultimate equality follows since $y \odot 1 \geq y$, $x \odot y \odot 1 \geq x \odot y \oslash 1$ and $x \geq x \oslash 1 \,\forall\, x, y$. Therefore, the tropical hypersurface $\mathcal{T}(f)$ is defined as the of $(x, y)$ where $f$ achieves its maximum at least twice in its monomials. That is to say,

$$
\begin{aligned}
\mathcal{T}(f) = &\{f(x, y) = (x^2 \oslash 1) = (x \odot y \odot 1)\} \cup \{f(x, y) = x^2 \oslash 1 = x\} \cup \\
&\{(f(x, y) = x = 0\} \cup \{f(x, y) = x = x \odot y \odot 1\} \cup \\
&\{f(x, y) = y \odot 1 = 0\} \cup \{f(x, y) = y \odot 1 = x \odot y \odot 1\} \cup \\
&\{f(x, y) = y \odot 1 = y^2 \odot 1\} \cup \{f(x, y) = y^2 \odot 1 = x \odot y \odot 1\}.
\end{aligned}
$$

This set $\mathcal{T}(f)$ is shown by the red lines in the second example in Figure 1. As for constructing the dual subdivision $\delta(f)$, we project the upperfaces in the newton polygon $\mathcal{P}(f)$ to $\mathbb{R}^2$. Note that $\mathcal{P}(f)$ with biases as per the definition in Section 2 is given as $\mathcal{P}(f) = \text{ConvHull}\{(\mathbf{a}_i, c_i) \in \mathbb{R}^2 \times \mathbb{R} \,\forall i = 1, \ldots, 6\}$ where $(\mathbf{a}_i, c_i)$ are the exponents and biases in the monomials of $f$, respectively. Therefore, $\mathcal{P}(f) = \text{ConvHull}\{(2, 0, -1), (1, 1, 1), (1, 0, 0), (0, 2, 1), (0, 1, 1), (0, 0, 0)\}$ as shown in Figure 6(a). As per Definition 7, the set of upper faces of $\mathcal{P}$ is:

$$
\begin{aligned}
\text{UP}(\mathcal{P}(f)) = \ &\text{ConvHull}\{(0, 2, 1), (1, 1, 1), (0, 1, 1)\} \cup \text{ConvHull}\{(0, 1, 1), (1, 1, 1), (1, 0, 0)\} \\
&\cup \text{ConvHull}\{(0, 1, 1), (1, 0, 0), (0, 0, 0)\} \cup \text{ConvHull}\{(1, 1, 1), (2, 0, -1), (1, 0, 0)\}.
\end{aligned}
$$

This set $\text{UP}(\mathcal{P}(f))$ is then projected, through $\pi$, to $\mathbb{R}^2$ shown in the yellow dashed lines in Figure 6(a) to construct the dual subdivision $\delta(f)$ in Figure 6(b). For example, note that the point $(0, 2, 1) \in \text{UF}(f)$ and thereafter, $\pi(0, 2, 1) = (0, 2, 0) \in \delta(f)$.

## 10 PROOF OF THEOREM 2

**Theorem 2.** *For a bias-free neural network in the form of $f(\mathbf{x}) : \mathbb{R}^n \to \mathbb{R}^2$ where $\mathbf{A} \in \mathbb{Z}^{p \times n}$ and $\mathbf{B} \in \mathbb{Z}^{2 \times p}$, let $R(\mathbf{x}) = H_1(\mathbf{x}) \odot Q_2(\mathbf{x}) \oplus H_2(\mathbf{x}) \odot Q_1(\mathbf{x})$ be a tropical polynomial. Then:*

- *Let $\mathcal{B} = \{\mathbf{x} \in \mathbb{R}^n : f_1(\mathbf{x}) = f_2(\mathbf{x})\}$ define the decision boundaries of $f$, then $\mathcal{B} \subseteq \mathcal{T}(R(\mathbf{x}))$.*

- *$\delta(R(\mathbf{x})) = \text{ConvHull}(\mathcal{Z}_{\mathbf{G}_1}, \mathcal{Z}_{\mathbf{G}_2})$. $\mathcal{Z}_{\mathbf{G}_1}$ is a zonotope in $\mathbb{R}^n$ with line segments $\{(\mathbf{B}^+(1, j) + \mathbf{B}^-(2, j))[\mathbf{A}^+(j, :), \mathbf{A}^-(j, :)]\}_{j=1}^p$ and shift $(\mathbf{B}^-(1, :) + \mathbf{B}^+(2, :))\mathbf{A}^-$. $\mathcal{Z}_{\mathbf{G}_2}$ is a zonotope in $\mathbb{R}^n$ with line segments $\{(\mathbf{B}^-(1, j) + \mathbf{B}^+(2, j))[\mathbf{A}^+(j, :), \mathbf{A}^-(j, :)]\}_{j=1}^p$ and shift $(\mathbf{B}^+(1, :) + \mathbf{B}^-(2, :))\mathbf{A}^-$. The line segment $(\mathbf{B}^+(1, j) + \mathbf{B}^-(2, j))[\mathbf{A}^+(j, :), \mathbf{A}^-(j, :)]$ has end points $\mathbf{A}^+(j, :)$ and $\mathbf{A}^-(j, :)$ in $\mathbb{R}^n$ and scaled by $(\mathbf{B}^+(1, j) + \mathbf{B}^-(2, j))$.*

*Note that $\mathbf{A}^+ = \max(\mathbf{A}, 0)$ and $\mathbf{A}^- = \max(-\mathbf{A}, 0)$ where the $\max(.)$ is element-wise. The line segment $(\mathbf{B}(1, j)^+ + \mathbf{B}(2, j)^-)[\mathbf{A}(j, :)^+, \mathbf{A}(j, :)^-]$ is one that has the end points $\mathbf{A}(j, :)^+$ and $\mathbf{A}(j, :)^-$ in $\mathbb{R}^n$ and scaled by the constant $\mathbf{B}(1, j)^+ + \mathbf{B}(2, j)^-$.*

*Proof.* For the first part, recall from Theorem 1 that both $f_1$ and $f_2$ are tropical rationals and hence,

$$
f_1(\mathbf{x}) = H_1(\mathbf{x}) - Q_1(\mathbf{x}) \qquad f_2(\mathbf{x}) = H_2(\mathbf{x}) - Q_2(\mathbf{x})
$$

Thus

$$
\begin{aligned}
\mathcal{B} &= \{x \in \mathbb{R}^n : f_1(\mathbf{x}) = f_2(\mathbf{x})\} = \{x \in \mathbb{R}^n : H_1(\mathbf{x}) - Q_1(\mathbf{x}) = H_2(\mathbf{x}) - Q_2(\mathbf{x})\} \\
&= \{x \in \mathbb{R}^n : H_1(\mathbf{x}) + Q_2(\mathbf{x}) = H_2(\mathbf{x}) + Q_1(\mathbf{x})\} \\
&= \{x \in \mathbb{R}^n : H_1(\mathbf{x}) \odot Q_2(\mathbf{x}) = H_2(\mathbf{x}) \odot Q_1(\mathbf{x})\}
\end{aligned}
$$

Recall that the tropical hypersurface is defined as the set of $\mathbf{x}$ where the maximum is attained by two or more monomials. Therefore, the tropical hypersurface of $R(\mathbf{x})$ is the set of $\mathbf{x}$ where the maximum is attained by two or more monomials in $(H_1(\mathbf{x}) \odot Q_2(\mathbf{x}))$, or attained by two or more monomials in

$(H_2(\mathbf{x}) \odot Q_1(\mathbf{x}))$, or attained by monomials in both of them in the same time, which is the decision boundaries. Hence, we can rewrite that as

$$\mathcal{T}(R(\mathbf{x})) = \mathcal{T}(H_1(\mathbf{x}) \odot Q_2(\mathbf{x})) \cup \mathcal{T}(H_2(\mathbf{x}) \odot Q_1(\mathbf{x})) \cup \mathcal{B}.$$

Therefore $\mathcal{B} \subseteq \mathcal{T}(R(x))$. For the second part of the Theorem, we first use the decomposition proposed by Zhang et al. (2018); Berrada et al. (2016) to show that for a network $f(\mathbf{x}) = \mathbf{B}\max(\mathbf{Ax}, \mathbf{0})$, it can be decomposed as tropical rational as follows

$$f(\mathbf{x}) = \left(\mathbf{B}^+ - \mathbf{B}^-\right)\left(\max(\mathbf{A}^+\mathbf{x}, \mathbf{A}^-\mathbf{x}) - \mathbf{A}^-\mathbf{x}\right)$$

$$= \left[\mathbf{B}^+ \max(\mathbf{A}^+\mathbf{x}, \mathbf{A}^-\mathbf{x}) + \mathbf{B}^-\mathbf{A}^-\mathbf{x}\right]$$

$$- \left[\mathbf{B}^- \max(\mathbf{A}^+\mathbf{x}, \mathbf{A}^-\mathbf{x}) + \mathbf{B}^+\mathbf{A}^-\mathbf{x}\right].$$

Therefore, we have that

$$H_1(\mathbf{x}) + Q_2(\mathbf{x}) = \left(\mathbf{B}^+(1,:) + \mathbf{B}^-(2,:)\right)\max(\mathbf{A}^+\mathbf{x}, \mathbf{A}^-\mathbf{x})$$

$$+ \left(\mathbf{B}^-(1,:) + \mathbf{B}^+(2,:)\right)\mathbf{A}^-\mathbf{x}$$

$$H_2(\mathbf{x}) + Q_1(\mathbf{x}) = \left(\mathbf{B}^-(1,:) + \mathbf{B}^+(2,:)\right)\max(\mathbf{A}^+\mathbf{x}, \mathbf{A}^-\mathbf{x})$$

$$+ \left(\mathbf{B}^+(1,:) + \mathbf{B}^-(2,:)\right)\mathbf{A}^-\mathbf{x}.$$

Therefore, note that:

$$\delta(R(\mathbf{x})) = \delta\left(\left(H_1(\mathbf{x}) \odot Q_2(\mathbf{x})\right) \oplus \left(H_2(\mathbf{x}) \odot Q_1(\mathbf{x})\right)\right)$$

$$\overset{4}{=} \text{ConvexHull}\left(\delta\left(H_1(\mathbf{x}) \odot Q_2(\mathbf{x})\right), \delta\left(H_2(\mathbf{x}) \odot Q_1(\mathbf{x})\right)\right)$$

$$\overset{3}{=} \text{ConvexHull}\left(\delta\left(H_1(\mathbf{x})\right)\tilde{+}\delta\left(Q_2(\mathbf{x})\right), \delta\left(H_2(\mathbf{x})\right)\tilde{+}\delta\left(Q_1(\mathbf{x})\right)\right).$$

Now observe that $H_1(\mathbf{x}) = \sum_{j=1}^{p}\left(\mathbf{B}^+(1,j) + \mathbf{B}^-(2,j)\right)\max\left(\mathbf{A}^+(j,:), \mathbf{A}^-(j,:)\mathbf{x}\right)$ tropically is given as follows $H_1(\mathbf{x}) = \odot_{j=1}^{p}\left[\mathbf{x}^{\mathbf{A}^+(j,:)} \oplus \mathbf{x}^{\mathbf{A}^-(j,:)}\right]^{\mathbf{B}^+(1,j)\odot\mathbf{B}^-(2,j)}$, thus we have that :

$$\delta(H_1(\mathbf{x})) = \left(\mathbf{B}^+(1,1) + \mathbf{B}^-(2,1)\right)\delta\left(\mathbf{x}^{\mathbf{A}^+(1,:)} \oplus \mathbf{x}^{\mathbf{A}^-(1,:)}\right)\tilde{+}\dots$$

$$\tilde{+}\left(\mathbf{B}^+(1,p) + \mathbf{B}^-(2,p)\right)\left(\delta(\mathbf{x}^{\mathbf{A}^+(p,:)} \oplus \mathbf{x}^{\mathbf{A}^-(p,:)})\right)$$

$$= \left(\mathbf{B}^+(1,1) + \mathbf{B}^-(2,1)\right)\text{ConvexHull}\left(\mathbf{A}^+(1,:), \mathbf{A}^-(1,:)\right)\tilde{+}\dots$$

$$\tilde{+}\left(\mathbf{B}^+(1,p) + \mathbf{B}^-(2,p)\right)\text{ConvexHull}\left(\mathbf{A}^+(p,:), \mathbf{A}^-(p,:)\right).$$

The operator $\tilde{+}$ indicates a Minkowski sum between sets. Note that $\text{ConvexHull}\left(\mathbf{A}^+(i,:), \mathbf{A}^-(i,:)\right)$ is the convexhull between two points which is a line segment in $\mathbb{Z}^n$ with end points that are $\{\mathbf{A}^+(i,:), \mathbf{A}^-(i,:)\}$ scaled with $\mathbf{B}^+(1,i) + \mathbf{B}^-(2,i)$. Observe that $\delta(F_1(\mathbf{x}))$ is a Minkowski sum of line segments which is is a zonotope. Moreover, note that $Q_2(\mathbf{x}) = (\mathbf{B}^-(1,:) + \mathbf{B}^+(2,:))\mathbf{A}^-\mathbf{x}$ tropically is given as follows $Q_2(\mathbf{x}) = \odot_{j=1}^{p}\mathbf{x}^{\mathbf{A}^-(j,:)(\mathbf{B}^+(1,j)\odot\mathbf{B}^-(2,j))}$. One can see that $\delta(Q_2(\mathbf{x}))$ is the Minkowski sum of the points $\{(\mathbf{B}^-(1,j) - \mathbf{B}^+(2,j))\mathbf{A}^-(j,:)\}\forall j$ in $\mathbb{R}^n$ (which is a standard sum) resulting in a point. Lastly, $\delta(H_1(\mathbf{x}))\tilde{+}\delta(Q_2(\mathbf{x}))$ is a Minkowski sum between a zonotope and a single point which corresponds to a shifted zonotope. A similar symmetric argument can be applied for the second part $\delta(H_2(\mathbf{x}))\tilde{+}\delta(Q_1(\mathbf{x}))$. $\square$

It is also worthy to mention that the extension to network with multi class output is trivial. In that case all of the analysis can be exactly applied studying the decision boundary between any two classes $(i, j)$ where $\mathcal{B} = \{x \in \mathbb{R}^n : f_i(\mathbf{x}) = f_j(\mathbf{x})\}$ and the rest of the proof will be exactly the same.

## 11 PROOF OF PROPOSITION 1

**Proposition 1.** *The zonotope formed by $p$ line segments in $\mathbb{R}^n$ with arbitrary end points $\{[\mathbf{u}_1^i, \mathbf{u}_2^i]\}_{i=1}^p$ is equivalent to the zonotope formed by the line segments $\{[\mathbf{u}_1^i - \mathbf{u}_2^i, \mathbf{0}]\}_{i=1}^p$ with a shift of $\sum_{i=1}^p \mathbf{u}_2^i$.*

*Proof.* Let $\mathbf{U}_j$ be a matrix with $\mathbf{U}_j(:, i) = \mathbf{u}_j^i, i = 1, \ldots, p$, $\mathbf{w}$ be a column-vector with $\mathbf{w}(i) = w_i, i = 1, \ldots, p$ and $\mathbf{1}_p$ is a column-vector of ones of length $p$. Then, the zonotope $\mathcal{Z}$ formed by the Minkowski sum of line segments with arbitrary end points can be defined as:

$$
\begin{aligned}
\mathcal{Z} &= \Big\{ \sum_{i=1}^p w_i \mathbf{u}_1^i + (1 - w_i)\mathbf{u}_2^i; w_i \in [0, 1], \ \forall \ i \Big\} \\
&= \Big\{ \mathbf{U}_1 \mathbf{w} - \mathbf{U}_2 \mathbf{w} + \mathbf{U}_2 \mathbf{1}_p, \ \ \mathbf{w} \in [0, 1]^p \Big\} \\
&= \Big\{ (\mathbf{U}_1 - \mathbf{U}_2) \mathbf{w} + \mathbf{U}_2 \mathbf{1}_p, \ \ \mathbf{w} \in [0, 1]^p \Big\} \\
&= \Big\{ (\mathbf{U}_1 - \mathbf{U}_2) \mathbf{w}, \ \ \mathbf{w} \in [0, 1]^p \Big\} \tilde{+} \Big\{ \mathbf{U}_2 \mathbf{1}_p \Big\}.
\end{aligned}
$$

Since the Minkowski sum of between a polytope and a point is a translation; thereafter, the proposition follows directly from Definition 6. $\qquad\square$

**Corollary 2.** *The generators of $\mathcal{Z}_{\mathbf{G}_1}, \mathcal{Z}_{\mathbf{G}_2}$ in Theorem 2 can be defined as $\mathbf{G}_1 = Diag[(\mathbf{B}^+(1,:)) + (\mathbf{B}^-(2,:))]\mathbf{A}$ and $\mathbf{G}_2 = Diag[(\mathbf{B}^+(2,:)) + (\mathbf{B}^-(1,:))]\mathbf{A}$, both with shift $(\mathbf{B}^-(1,:) + \mathbf{B}^+(2,:) + \mathbf{B}^+(1,:) + \mathbf{B}^-(2,:)) \mathbf{A}^-$, where $Diag(\mathbf{v})$ arranges $\mathbf{v}$ in a diagonal matrix.*

*Proof.* This follows directly by applying Proposition 1 to the second bullet point of Theorem 2. $\quad\square$

### 11.1 OPTIMIZATION OF OBJECTIVE 2 OF THE BINARY CLASSIFIER

$$
\min_{\tilde{\mathbf{A}}, \tilde{\mathbf{B}}} \frac{1}{2} \left\| \tilde{\mathbf{G}}_1 - \mathbf{G}_1 \right\|_F^2 + \left\| \frac{1}{2} \tilde{\mathbf{G}}_2 - \mathbf{G}_2 \right\|_F^2 + \lambda_1 \left\| \tilde{\mathbf{G}}_1 \right\|_{2,1} + \lambda_2 \left\| \tilde{\mathbf{G}}_2 \right\|_{2,1}. \tag{5}
$$

Note that $\tilde{\mathbf{G}}_1 = \text{Diag}\Big[\text{ReLU}(\tilde{\mathbf{B}}(1,:)) + \text{ReLU}(-\tilde{\mathbf{B}}(2,:))\Big]\tilde{\mathbf{A}}$, $\tilde{\mathbf{G}}_2 = \text{Diag}\Big[\text{ReLU}(\tilde{\mathbf{B}}(2,:)) + \text{ReLU}(-\tilde{\mathbf{B}}(1,:))\Big]\tilde{\mathbf{A}}$. Note that $\mathbf{G}_1 = \text{Diag}\Big[\text{ReLU}(\mathbf{B}(1,:)) + \text{ReLU}(-\mathbf{B}(2,:))\Big]\mathbf{A}$ and $\mathbf{G}_2 = \text{Diag}\Big[\text{ReLU}(\mathbf{B}(2,:)) + \text{ReLU}(-\mathbf{B}(1,:))\Big]\mathbf{A}$. For ease of notation, we refer to $\text{ReLU}(\tilde{\mathbf{B}}(i,:))$ and $\text{ReLU}(-\tilde{\mathbf{B}}(i,:))$ as $\tilde{\mathbf{B}}^+(i,:)$ and $\tilde{\mathbf{B}}^-(i,:)$, respectively. We solve the problem with co-ordinate descent an alternate over variables.

**Update $\tilde{\mathbf{A}}$.**

$$
\tilde{\mathbf{A}} \leftarrow \arg\min_{\tilde{\mathbf{A}}} \frac{1}{2} \left\| \text{Diag}(\mathbf{c}_1) \tilde{\mathbf{A}} - \mathbf{G}_1 \right\|_F^2 + \frac{1}{2} \left\| \text{Diag}(\mathbf{c}_2)\tilde{\mathbf{A}} - \mathbf{G}_2 \right\|_F^2 + \lambda_1 \left\| \text{Diag}(\mathbf{c}_1)\tilde{\mathbf{A}} \right\|_{2,1} + \lambda_2 \left\| \text{Diag}(\mathbf{c}_2)\tilde{\mathbf{A}} \right\|_{2,1},
$$

where $\mathbf{c}_1 = \text{ReLU}(\mathbf{B}(1,:)) + \text{ReLU}(-\mathbf{B}(2,:))$ and $\mathbf{c}_2 = \text{ReLU}(\mathbf{B}(2,:)) + \text{ReLU}(-\mathbf{B}(1,:))$. Note that the problem is separable per-row of $\tilde{\mathbf{A}}$. Therefore, the problem reduces to updating rows of $\tilde{\mathbf{A}}$ independently and the problem exhibits a closed form solution.

$$\tilde{\mathbf{A}}(i,:) = \underset{\tilde{\mathbf{A}}(i,:)}{\arg\min} \frac{1}{2}\left\|\mathbf{c}_1^i\tilde{\mathbf{A}}(i,:) - \mathbf{G}_1(i,:)\right\|_2^2 + \frac{1}{2}\left\|\mathbf{c}_2^i\tilde{\mathbf{A}}(i,:) - \mathbf{G}_2(i,:)\right\|_2^2 + \left(\lambda_1\sqrt{\mathbf{c}_1^i} + \lambda_2\sqrt{\mathbf{c}_2^i}\right)\left\|\tilde{\mathbf{A}}(i,:)\right\|_2$$

$$= \underset{\tilde{\mathbf{A}}(i,:)}{\arg\min} \frac{1}{2}\left\|\tilde{\mathbf{A}}(i,:) - \frac{\mathbf{c}_1^i\mathbf{G}_1(i,:) + \mathbf{c}_2^i\mathbf{G}_2(i,:)}{\frac{1}{2}(\mathbf{c}_1^i + \mathbf{c}_2^i)}\right\|_2^2 + \frac{1}{2}\frac{\lambda_1\sqrt{\mathbf{c}_1^i} + \lambda_2\sqrt{\mathbf{c}_2^i}}{\frac{1}{2}(\mathbf{c}_1^i + \mathbf{c}_2^i)}\left\|\tilde{\mathbf{A}}(i,:)\right\|_2$$

$$= \max\left(1 - \frac{1}{2}\frac{\lambda_1\sqrt{\mathbf{c}_1^i} + \lambda_2\sqrt{\mathbf{c}_2^i}}{\frac{1}{2}(\mathbf{c}_1^i + \mathbf{c}_2^i)}\frac{1}{\left\|\frac{\mathbf{c}_1^i\mathbf{G}_1(i,:) + \mathbf{c}_2^i\mathbf{G}_2(i,:)}{\frac{1}{2}(\mathbf{c}_1^i + \mathbf{c}_2^i)}\right\|_2}, 0\right)\left(\frac{\mathbf{c}_1^i\mathbf{G}_1(i,:) + \mathbf{c}_2^i\mathbf{G}_2(i,:)}{\frac{1}{2}(\mathbf{c}_1^i + \mathbf{c}_2^i)}\right).$$

**Update $\tilde{\mathbf{B}}^+(1,:)$.**

$$\tilde{\mathbf{B}}^+(1,:) = \underset{\tilde{\mathbf{B}}^+(1,:)}{\arg\min} \frac{1}{2}\left\|\text{Diag}\left(\tilde{\mathbf{B}}^+(1,:)\right)\tilde{\mathbf{A}} - \mathbf{C}_1\right\|_F^2 + \lambda_1\left\|\text{Diag}\left(\tilde{\mathbf{B}}^+(1,:)\right)\tilde{\mathbf{A}} + \mathbf{C}_2\right\|_{2,1}, \quad \text{s.t. } \tilde{\mathbf{B}}^+(1,:) \geq \mathbf{0}.$$

Note that $\mathbf{C}_1 = \mathbf{G}_1 - \text{Diag}\left(\tilde{\mathbf{B}}^-(2,:)\right)\tilde{\mathbf{A}}$ and where $\text{Diag}\left(\tilde{\mathbf{B}}^-(2,:)\right)\tilde{\mathbf{A}}$. Note the problem is separable in the coordinates of $\tilde{\mathbf{B}}^+(1,:)$ and a projected gradient descent can be used to solve the problem in such a way as:

$$\tilde{\mathbf{B}}^+(1,j) = \underset{\tilde{\mathbf{B}}^+(1,j)}{\arg\min} \frac{1}{2}\left\|\tilde{\mathbf{B}}^+(1,j)\tilde{\mathbf{A}}(j,:) - \mathbf{C}_1(j,:)\right\|_2^2 + \lambda_1\left\|\tilde{\mathbf{B}}^+(1,j)\tilde{\mathbf{A}}(j,:) + \mathbf{C}_2(j,:)\right\|_2, \quad \text{s.t. } \tilde{\mathbf{B}}^+(1,j) \geq 0.$$

A similar symmetric argument can be used to update the variables $\tilde{\mathbf{B}}^+(2,:)$, $\tilde{\mathbf{B}}^+(1,:)$ and $\tilde{\mathbf{B}}^-(2,:)$.

## 12 ADAPTING OPTIMIZATION 2 FOR MULTI-CLASS CLASSIFIER

Note that Theorem 2 describes a superset to the decision boundaries of a binary classifier through the dual subdivision $R(\mathbf{x})$, *i.e.* $\delta(R(\mathbf{x}))$. For a neural network $f$ with $k$ classes, a natural extension for it is to analyze the pair-wise decision boundaries of of all $k$-classes. Thus, let $\mathcal{T}(R_{ij}(\mathbf{x}))$ be the superset to the decision boundaries separating classes $i$ and $j$. Therefore, a natural extension to the geometric loss in Equation 1 is to preserve the polytopes among all pairwise follows:

$$\min_{\tilde{\mathbf{A}},\tilde{\mathbf{B}}} \sum_{\forall[i,j]\in S} d\Big(\text{ConvexHull}\left(\mathcal{Z}_{\tilde{\mathbf{G}}_{(i+,j-)}}, \mathcal{Z}_{\tilde{\mathbf{G}}_{(j+,i-)}}\right), \text{ConvexHull}\left(\mathcal{Z}_{\mathbf{G}_{(i+,j-)}}, \mathcal{Z}_{\mathbf{G}_{(j+,i-)}}\right)\Big). \quad (6)$$

The set $S$ is all possible pairwise combinations of the $k$ classes such that $S = \{\{i,j\}, \forall i \neq j, i = 1,\ldots,k, j = 1,\ldots,k\}$. The generator $\mathcal{Z}(\tilde{G}_{(i,j)})$ is the zonotope with the generator matrix $\tilde{\mathbf{G}}_{(i+,j-)} = \text{Diag}\left[\text{ReLU}(\tilde{\mathbf{B}}(i,:)) + \text{ReLU}(-\tilde{\mathbf{B}}(j,:))\right]\tilde{\mathbf{A}}$. However, such an approach is generally computationally expensive, particularly, when $k$ is very large. To this end, we make the following observation that $\tilde{\mathbf{G}}_{(i+,j-)}$ can be equivalently written as a Minkowski sum between two sets zonotopes with the generators $\mathbf{G}_{i+} = \text{Diag}\left[\text{ReLU}(\tilde{\mathbf{B}}(i,:))\right]\tilde{\mathbf{A}}$ and $\mathbf{G}_{j-} = \text{Diag}\left[\text{ReLU}(\tilde{\mathbf{B}}_{j-})\right]\tilde{\mathbf{A}}$. That is to say, $\mathcal{Z}_{\tilde{\mathbf{G}}_{(i+,j-)}} = \mathcal{Z}_{\tilde{\mathbf{G}}_{i+}}\tilde{+}\mathcal{Z}_{\tilde{\mathbf{G}}_{j-}}$. This follows from the associative property of Minkowski sums given as follows:

**Fact 5.** *Let $\{S_i\}_{i=1}^n$ be the set of $n$ line segments. Then we have that*

$$S = S_1\tilde{+}\ldots\tilde{+}S_n = P\tilde{+}V$$

*where the sets $P = \tilde{+}_{j\in C_1}S_j$ and $V = \tilde{+}_{j\in C_2}S_j$ where $C_1$ and $C_2$ are any complementary partitions of the set $\{S_i\}_{i=1}^n$.*

Hence, $\tilde{\mathbf{G}}_{(i+,j-)}$ can be seen a concatenation between $\tilde{\mathbf{G}}_{i+}$ and $\tilde{\mathbf{G}}_{j-}$. Thus, the objective in 6 can be expanded as follows:

$$\min_{\tilde{\mathbf{A}},\tilde{\mathbf{B}}} \sum_{\forall\{i,j\}\in S} d\Big(\text{ConvexHull}\left(\mathcal{Z}_{\tilde{\mathbf{G}}_{(i+,j-)}}, \mathcal{Z}_{\tilde{\mathbf{G}}_{(j+,i-)}}\right), \text{ConvexHull}\left(\mathcal{Z}_{\mathbf{G}_{(i+,j-)}}, \mathcal{Z}_{\mathbf{G}_{(j+,i-)}}\right)\Big)$$

$$= \min_{\tilde{\mathbf{A}},\tilde{\mathbf{B}}} \sum_{\forall\{i,j\}\in S} d\Big(\text{ConvexHull}\left(\mathcal{Z}_{\tilde{\mathbf{G}}_{i+}}\tilde{+}\mathcal{Z}_{\tilde{\mathbf{G}}_{j-}}, \mathcal{Z}_{\tilde{\mathbf{G}}_{j}^+}\tilde{+}\mathcal{Z}_{\tilde{\mathbf{G}}_{i-}}\right), \text{ConvexHull}\left(\mathcal{Z}_{\mathbf{G}_{i+}}\tilde{+}\mathcal{Z}_{\mathbf{G}_{j-}}, \mathcal{Z}_{\mathbf{G}_{j}^+}\tilde{+}\mathcal{Z}_{\mathbf{G}_{i-}}\right)\Big)$$

$$\approx \min_{\tilde{\mathbf{A}},\tilde{\mathbf{B}}} \sum_{\forall[i,j]\in S} \left\| \begin{pmatrix}\tilde{\mathbf{G}}_{i+}\\\tilde{\mathbf{G}}_{j-}\end{pmatrix} - \begin{pmatrix}\mathbf{G}_{i+}\\\mathbf{G}_{j-}\end{pmatrix} \right\|_F^2 + \left\| \begin{pmatrix}\tilde{\mathbf{G}}_{i-}\\\tilde{\mathbf{G}}_{j+}\end{pmatrix} - \begin{pmatrix}\mathbf{G}_{i-}\\\mathbf{G}_{j+}\end{pmatrix} \right\|_F^2$$

$$= \min_{\tilde{\mathbf{A}},\tilde{\mathbf{B}}} \sum_{\forall\{i,j\}\in S} \frac{1}{2}\left\|\tilde{\mathbf{G}}_{i+} - \mathbf{G}_{i+}\right\|_F^2 + \frac{1}{2}\left\|\tilde{\mathbf{G}}_{i-} - \mathbf{G}_{i-}\right\|_F^2 + \frac{1}{2}\left\|\tilde{\mathbf{G}}_{j+} - \mathbf{G}_{j+}\right\|_F^2 + \frac{1}{2}\left\|\tilde{\mathbf{G}}_{j-} - \mathbf{G}_{j-}\right\|_F^2$$

$$= \min_{\tilde{\mathbf{A}},\tilde{\mathbf{B}}} \frac{k-1}{2}\sum_{i=1}^{k} \left\|\tilde{\mathbf{G}}_{i+} - \mathbf{G}_{i+}\right\|_F^2 + \left\|\tilde{\mathbf{G}}_{i-} - \mathbf{G}_{i-}\right\|_F^2.$$

The approximation follows in a similar argument to the binary classifier case. The last equality follows from a counting argument. We solve the objective for all multi-class networks in the experiments with alternating optimization in a similar fashion to the binary classifier case. Similarly to the binary classification approach, we introduce the $\|.\|_{2,1}$ to enforce sparsity constraints for pruning purposes. Therefore the overall objective has the form:

$$\min_{\tilde{\mathbf{A}},\tilde{\mathbf{B}}} \frac{1}{2}\sum_{i=1}^{k} \left\|\tilde{\mathbf{G}}_{i+} - \mathbf{G}_{i+}\right\|_F^2 + \left\|\tilde{\mathbf{G}}_{i-} - \mathbf{G}_{i-}\right\|_F^2 + \lambda\left(\left\|\tilde{\mathbf{G}}_{i+}\right\|_{2,1} + \left\|\tilde{\mathbf{G}}_{i-}\right\|_{2,1}\right).$$

For completion, we derive the updates for $\tilde{\mathbf{A}}$ and $\tilde{\mathbf{B}}$.

**Update $\tilde{\mathbf{A}}$.**

$$\tilde{\mathbf{A}} = \arg\min_{\tilde{\mathbf{A}}} \sum_{i=1}^{k} \frac{1}{2}\left(\left\|\text{Diag}\left(\tilde{\mathbf{B}}^+(i,:)\right)\tilde{\mathbf{A}} - \mathbf{G}_{i+}\right\|_F^2 + \left\|\text{Diag}\left(\tilde{\mathbf{B}}^-(i,:)\right)\tilde{\mathbf{A}} - \mathbf{G}_{i-}\right\|_F^2\right)$$

$$+ \lambda\left(\left\|\text{Diag}\left(\tilde{\mathbf{B}}^+(i,:)\right)\tilde{\mathbf{A}}\right\|_{2,1} + \left\|\text{Diag}\left(\tilde{\mathbf{B}}^-(i,:)\right)\tilde{\mathbf{A}}\right\|_{2,1}\right).$$

Similar to the binary classification, the problem is separable in the rows of $\tilde{\mathbf{A}}$. and a closed form solution in terms of the proximal operator of $\ell_2$ norm follows naturally for each $\tilde{\mathbf{A}}(i,:)$.

**Update $\tilde{\mathbf{B}}^+(i,:)$.**

$$\tilde{\mathbf{B}}^+(i,:) = \arg\min_{\tilde{\mathbf{B}}^+(i,:)} \frac{1}{2}\left\|\text{Diag}\left(\tilde{\mathbf{B}}^+(i,:)\right)\tilde{\mathbf{A}} - \tilde{\mathbf{G}}_{i+}\right\|_F^2 + \lambda\left\|\text{Diag}\left(\tilde{\mathbf{B}}^+(i,:)\right)\tilde{\mathbf{A}}\right\|_{2,1}, \text{ s.t. } \tilde{\mathbf{B}}^+(i,:) \geq \mathbf{0}.$$

Note that the problem is separable per coordinates of $\mathbf{B}^+(i,:)$ and each subproblem is updated as:

$$\tilde{\mathbf{B}}^+(i,j) = \arg\min_{\tilde{\mathbf{B}}^+(i,j)} \frac{1}{2}\left\|\tilde{\mathbf{B}}^+(i,j)\tilde{\mathbf{A}}(j,:) - \tilde{\mathbf{G}}_{i+}(j,:)\right\|_2^2 + \lambda\left\|\tilde{\mathbf{B}}^+(i,j)\tilde{\mathbf{A}}(j,:)\right\|_2, \text{ s.t. } \tilde{\mathbf{B}}^+(i,j) \geq 0$$

$$= \arg\min_{\tilde{\mathbf{B}}^+(i,j)} \frac{1}{2}\left\|\tilde{\mathbf{B}}^+(i,j)\tilde{\mathbf{A}}(j,:) - \tilde{\mathbf{G}}_{i+}(j,:)\right\|_2^2 + \lambda\left|\tilde{\mathbf{B}}(i,j)\right|\left\|\tilde{\mathbf{A}}(j,:)\right\|_2, \text{ s.t. } \tilde{\mathbf{B}}^+(i,j) \geq 0$$

$$= \max\left(0, \frac{\tilde{\mathbf{A}}(j,:)^\top \tilde{\mathbf{G}}_{i+}(j,:) - \lambda\|\tilde{\mathbf{A}}(j,:)\|_2}{\|\tilde{\mathbf{A}}(j,:)\|_2^2}\right).$$

A similar argument can be used to update $\tilde{\mathbf{B}}^-(i,:) \forall i$. Finally, the parameters of the pruned network will be constructed $\mathbf{A} \leftarrow \tilde{\mathbf{A}}$ and $\mathbf{B} \leftarrow \tilde{\mathbf{B}}^+ - \tilde{\mathbf{B}}^-$.

## 13 TROPICAL ADVERSARIAL ATTACKS.

**Dual View to Adversarial Attacks.** For a classifier $f : \mathbb{R}^n \rightarrow \mathbb{R}^k$ and input $\mathbf{x}_0$ classified as $c$, a standard formulation for targeted adversarial attacks to a different class $t$ is defined as:

$$\min_\eta \ \mathcal{D}(\eta) \quad \text{s.t.} \quad \arg\max_i f_i(\mathbf{x}_0 + \eta) = t \neq c \tag{7}$$

This objective aims to compute the lowest energy input noise $\eta$ (measured by $\mathcal{D}$) such that the the new sample $(\mathbf{x}_0 + \eta)$ crosses the decision boundaries of $f$ to a new classification region. Here, we present a dual view to adversarial attacks. Instead of designing a sample noise $\eta$ such that $(\mathbf{x}_0 + \eta)$ belongs to a new decision region, one can instead fix $\mathbf{x}_0$ and perturb the network parameters to move the decision boundaries in a way that $\mathbf{x}_0$ appears in a new classification region. In particular, let $\mathbf{A}_1$ be the first linear layer of $f$, such that $f(\mathbf{x}_0) = g(\mathbf{A}_1\mathbf{x}_0)$. One can now perturb $\mathbf{A}_1$ to alter the decision boundaries and relate this parameter perturbation to the input perturbation as follows:

$$g((\mathbf{A}_1 + \xi_{\mathbf{A}_1})\mathbf{x}_0) = g(\mathbf{A}_1\mathbf{x}_0 + \xi_{\mathbf{A}_1}\mathbf{x}_0) = g(\mathbf{A}_1\mathbf{x}_0 + \mathbf{A}_1\eta) = f(\mathbf{x}_0 + \eta). \tag{8}$$

From this dual view, we observe that traditional adversarial attacks are intimately related to perturbing the parameters of the first linear layer through the linear system: $\mathbf{A}_1\eta = \xi_{\mathbf{A}_1}\mathbf{x}_0$. *The two views and formulations are identical* under such condition. With this analysis, Theorem 2 provides explicit means to geometrically construct adversarial attacks by perturbing the decision boundaries. In particular, since the normals to the dual subdivision polytope $\delta(R(\mathbf{x}))$ of a given DNN represent the tropical hypersurface $\mathcal{T}(R(\mathbf{x}))$, which is a superset to the decision boundaries set $\mathcal{B}$, $\xi_{\mathbf{A}_1}$ can be designed to sufficiently perturb the dual subdivision resulting in a change in the network prediction of $\mathbf{x}_0$ to the targeted class $t$. Based on this observation, we design an optimization problem that generates two sets of perturbations, an input perturbation and parameter perturbation, that are equivalent to each other.

**Formulation.** Based on this observation, we formulate the problem as follows:

$$\min_{\eta, \xi_{\mathbf{A}_1}} \quad \mathcal{D}_1(\eta) + \mathcal{D}_2(\xi_{\mathbf{A}_1}) \quad \text{s.t.} \quad -\text{loss}(g(\mathbf{A}_1(\mathbf{x}_0 + \eta)), t) \leq -1; \quad \|\eta\|_\infty \leq \epsilon_1;$$
$$-\text{loss}(g(\mathbf{A}_1 + \xi_{\mathbf{A}_1})\mathbf{x}_0, t) \leq -1; \quad (\mathbf{x}_0 + \eta) \in [0, 1]^n, \quad \|\xi_{\mathbf{A}_1}\|_{\infty,\infty} \leq \epsilon_2, \quad \mathbf{A}_1\eta = \xi_{\mathbf{A}_1}\mathbf{x}_0. \tag{9}$$

The loss is the standard cross-entropy loss. The first row of constraints ensures that the network prediction is the desired target class $t$ when the input $\mathbf{x}_0$ is perturbed by $\eta$, and equivalently by perturbing the first linear layer $\mathbf{A}_1$ by $\xi_{\mathbf{A}_1}$. This is identical to $f_1$ as proposed by Carlini & Wagner (2016). Moreover, the third and fourth constraints guarantee that the perturbed input is feasible and that the perturbation is bounded, respectively. The fifth constraint is to limit the maximum perturbation on the first linear layer, while the last constraint enforces the dual equivalence between input perturbation and parameter perturbation. The function $\mathcal{D}_2$ captures the perturbation of the dual subdivision polytope upon perturbing the first linear layer by $\xi_{\mathbf{A}_1}$. For a single hidden layer neural network parameterized as $(\mathbf{A}_1 + \xi_{\mathbf{A}_1}) \in \mathbb{R}^{p \times n}$ and $\mathbf{B} \in \mathbb{R}^{2 \times p}$ for the first and second layers respectively, $\mathcal{D}_2$ can capture the perturbations in each of the two zonotopes discussed in Theorem 2 and we define it as:

$$\mathcal{D}_2(\xi_{\mathbf{A}_1}) = \frac{1}{2} \sum_{j=1}^{2} \left\|\text{Diag}(\mathbf{B}^+(j,:))\xi_{\mathbf{A}_1}\right\|_F^2 + \left\|\text{Diag}(\mathbf{B}^-(j,:))\xi_{\mathbf{A}_1}\right\|_F^2. \tag{10}$$

We solve Problem (9) with a penalty method on the linear equality constraints, where each penalty step is solved with ADMM Boyd et al. (2011) in a similar fashion to the work of Xu et al. (2018).

The function $\mathcal{D}_2(\xi_{\mathbf{A}})$ captures the perturbation in the dual subdivision polytope such that the dual subdivision of the network with the first linear layer $\mathbf{A}_1$ is similar to the dual subdivision of the network with the first linear layer $\mathbf{A}_1 + \xi_{\mathbf{A}_1}$. This can be generally formulated as an approximation to the following distance function $d\left(\text{ConvHull}\left(\mathcal{Z}_{\tilde{\mathbf{G}}_1}, \mathcal{Z}_{\tilde{\mathbf{G}}_2}\right), \text{ConvHull}\left(\mathcal{Z}_{\mathbf{G}_1}, \mathcal{Z}_{\mathbf{G}_2}\right)\right)$, where $\tilde{\mathbf{G}}_1 = \text{Diag}\left[\text{ReLU}(\tilde{\mathbf{B}}(1,:)) + \text{ReLU}(-\tilde{\mathbf{B}}(2,:))\right]\left(\tilde{\mathbf{A}} + \xi_{\mathbf{A}_1}\right)$, $\tilde{\mathbf{G}}_2 = \text{Diag}\left[\text{ReLU}(\tilde{\mathbf{B}}(2,:)) + \text{ReLU}(-\tilde{\mathbf{B}}(1,:))\right]\left(\tilde{\mathbf{A}} + \xi_{\mathbf{A}_1}\right)$, $\mathbf{G}_1 = \text{Diag}\left[\text{ReLU}(\tilde{\mathbf{B}}(1,:)) + \text{ReLU}(-\tilde{\mathbf{B}}(2,:))\right]\tilde{\mathbf{A}}$ and $\mathbf{G}_2 = \text{Diag}\left[\text{ReLU}(\tilde{\mathbf{B}}(2,:)) + \text{ReLU}(-\tilde{\mathbf{B}}(1,:))\right]\tilde{\mathbf{A}}$. In particular, to approximate the function $d$, one can

use a similar argument as in used in network pruning 5 such that $\mathcal{D}_2$ approximates the generators of the zonotopes directly as follows:

$$
\begin{aligned}
\mathcal{D}_2(\xi_{\mathbf{A}_1}) &= \frac{1}{2}\left\|\tilde{\mathbf{G}}_1 - \mathbf{G}_1\right\|_F^2 + \frac{1}{2}\left\|\tilde{\mathbf{G}}_2 - \mathbf{G}_2\right\|_F^2 \\
&= \frac{1}{2}\left\|\text{Diag}\Big(\mathbf{B}^+(1,:)\Big)\xi_{\mathbf{A}_1}\right\|_F^2 + \frac{1}{2}\left\|\text{Diag}\Big(\mathbf{B}^-(1,:)\Big)\xi_{\mathbf{A}_1}\right\|_F^2 \\
&\quad + \frac{1}{2}\left\|\text{Diag}\Big(\mathbf{B}^+(2,:)\Big)\xi_{\mathbf{A}_1}\right\|_F^2 + \frac{1}{2}\left\|\text{Diag}\Big(\mathbf{B}^-(2,:)\Big)\xi_{\mathbf{A}_1}\right\|_F^2 .
\end{aligned}
$$

This can thereafter be extended to multi-class network with $k$ classes as follows $\mathcal{D}_2(\xi_{\mathbf{A}_1}) = \frac{1}{2}\sum_{j=1}^k \left\|\text{Diag}\Big(\mathbf{B}^+(j,:)\Big)\xi_{\mathbf{A}_1}\right\|_F^2 + \left\|\text{Diag}\Big(\mathbf{B}^-(j,:)\Big)\xi_{\mathbf{A}_1}\right\|_F^2$. Following Xu et al. (2018), we take $\mathcal{D}_1(\eta) = \frac{1}{2}\|\eta\|_2^2$. Therefore, we can write 9 as follows:

$$
\begin{aligned}
\min_{\eta,\xi_{\mathbf{A}}} \quad & \mathcal{D}_1(\eta) + \sum_{j=1}^k \left\|\text{Diag}\Big(\mathbf{B}^+(j,:)\Big)\xi_{\mathbf{A}}\right\|_F^2 + \left\|\text{Diag}\Big(\mathbf{B}^-(j,:)\Big)\xi_{\mathbf{A}}\right\|_F^2 . \\
\text{s.t.} \quad & -loss(g(\mathbf{A}_1(\mathbf{x}_0+\eta)),t) \leq -1, \quad -loss(g((\mathbf{A}_1+\xi_{\mathbf{A}_1})\mathbf{x}_0),t) \leq -1, \\
& (\mathbf{x}_0+\eta) \in [0,1]^n, \quad \|\eta\|_\infty \leq \epsilon_1, \quad \|\xi_{\mathbf{A}_1}\|_{\infty,\infty} \leq \epsilon_2, \quad \mathbf{A}_1\eta - \xi_{\mathbf{A}_1}\mathbf{x}_0 = 0.
\end{aligned}
$$

To enforce the linear equality constraints $\mathbf{A}_1\eta - \xi_{\mathbf{A}_1}\mathbf{x}_0 = 0$, we use a penalty method, where each iteration of the penalty method we solve the sub-problem with ADMM updates. That is, we solve the following optimization problem with ADMM with increasing $\lambda$ such that $\lambda \to \infty$. For ease of notation, lets denote $\mathcal{L}(\mathbf{x}_0+\eta) = -loss(g(\mathbf{A}_1(\mathbf{x}_0+\eta)),t)$, and $\bar{\mathcal{L}}(\mathbf{A}_1) = -loss(g((\mathbf{A}_1+\xi_{\mathbf{A}_1})\mathbf{x}_0),t)$.

$$
\begin{aligned}
\min_{\eta,z,w,\xi_{\mathbf{A}_1}} \quad & \|\eta\|_2^2 + \sum_{j=1}^k \left\|\text{Diag}\Big(\text{ReLU}(\mathbf{B}(j,:))\Big)\xi_{\mathbf{A}_1}\right\|_F^2 + \left\|\text{Diag}\Big(\text{ReLU}(-\mathbf{B}(j,:))\Big)\xi_{\mathbf{A}_1}\right\|_F^2 \\
& + \mathcal{L}(\mathbf{x}_0+\mathbf{z}) + h_1(\mathbf{w}) + h_2(\xi_{\mathbf{A}_1}) + \lambda\|\mathbf{A}_1\eta - \xi_{\mathbf{A}_1}\mathbf{x}_0\|_2^2 + \bar{\mathcal{L}}(\mathbf{A}_1). \\
\text{s.t.} \quad & \eta = \mathbf{z} \quad \mathbf{z} = \mathbf{w}.
\end{aligned}
$$

where

$$
h_1(\eta) = \begin{cases} 0, & \text{if } (\mathbf{x}_0+\eta)\in[0,1]^n, \|\eta\|_\infty \leq \epsilon_1 \\ \infty, & else \end{cases} \qquad h_2(\xi_{\mathbf{A}_1}) = \begin{cases} 0, & \text{if } \|\xi_{\mathbf{A}_1}\|_{\infty,\infty} \leq \epsilon_2 \\ \infty, & else \end{cases}.
$$

The augmented Lagrangian is given as follows:

$$
\begin{aligned}
\mathcal{L}(\eta,\mathbf{w},\mathbf{z},\xi_{\mathbf{A}_1},\mathbf{u},\mathbf{v}) := \quad & \|\eta\|_2^2 + \mathcal{L}(\mathbf{x}_0+\mathbf{z}) + h_1(\mathbf{w}) + \sum_{j=1}^k \left\|\text{Diag}(\mathbf{B}^+(j,:))\xi_{\mathbf{A}_1}\right\|_F^2 + \left\|\text{Diag}(\mathbf{B}^-(j,:))\xi_{\mathbf{A}_1}\right\|_F^2 \\
& + \bar{\mathcal{L}}(\mathbf{A}_1) + h_2(\xi_{\mathbf{A}_1}) + \lambda\|\mathbf{A}_1\eta - \xi_{\mathbf{A}_1}\mathbf{x}_0\|_2^2 + \mathbf{u}^\top(\eta-\mathbf{z}) + \mathbf{v}^\top(\mathbf{w}-\mathbf{z}) \\
& + \frac{\rho}{2}(\|\eta-\mathbf{z}\|_2^2 + \|\mathbf{w}-\mathbf{z}\|_2^2).
\end{aligned}
$$

Thereafter, ADMM updates are given as follows:

$$
\begin{aligned}
\{\eta^{k+1}, \mathbf{w}^{k+1}\} &= \arg\min_{\eta,\mathbf{w}} \mathcal{L}(\eta,\mathbf{w},\mathbf{z}^k,\xi_{\mathbf{A}_1}^k,\mathbf{u}^k,\mathbf{v}^k), \\
\mathbf{z}^{k+1} &= \arg\min_{\mathbf{z}} \mathcal{L}(\eta^{k+1},\mathbf{w}^{k+1},\mathbf{z},\xi_{\mathbf{A}_1}^k,\mathbf{u}^k,\mathbf{v}^k), \\
\xi_{\mathbf{A}_1}^{k+1} &= \arg\min_{\xi_{\mathbf{A}_1}} \mathcal{L}(\eta^{k+1},\mathbf{w}^{k+1},\mathbf{z}^{k+1},\xi_{\mathbf{A}_1},\mathbf{u}^k,\mathbf{v}^k).
\end{aligned}
$$

$$\mathbf{u}^{k+1} = \mathbf{u}^k + \rho(\eta^{k+1} - \mathbf{z}^{k+1}), \ \ \mathbf{v}^{k+1} = \mathbf{v}^k + \rho(\mathbf{w}^{k+1} - \mathbf{z}^{k+1}).$$

**Updating $\eta$:**

$$\eta^{k+1} = \arg\min_{\eta} \|\eta\|_2^2 + \lambda\|\mathbf{A}_1\eta - \xi_{\mathbf{A}_1}\mathbf{x}_0\|_2^2 + \mathbf{u}^\top\eta + \frac{\rho}{2}\|\eta - \mathbf{z}\|_2^2$$

$$= \left(2\lambda\mathbf{A}_1^\top\mathbf{A}_1 + (2+\rho)\mathbf{I}\right)^{-1}\left(2\lambda\mathbf{A}_1^\top\xi_{\mathbf{A}_1}^k\mathbf{x}_0 + \rho\mathbf{z}^k - \mathbf{u}^k\right).$$

**Updating $\mathbf{w}$:**

$$\mathbf{w}^{k+1} = \arg\min_{\mathbf{w}} \mathbf{v}^{k^\top}\mathbf{w} + h_1(\mathbf{w}) + \frac{\rho}{2}\|\mathbf{w} - \mathbf{z}^k\|_2^2$$

$$= \arg\min_{\mathbf{w}} \frac{1}{2}\left\|\mathbf{w} - \left(\mathbf{z}^k - \frac{\mathbf{v}^k}{\rho}\right)\right\|_2^2 + \frac{1}{\rho}h_1(\mathbf{w}).$$

The update $\mathbf{w}$ is separable in coordinates as follows:

$$\mathbf{w}^{k+1} = \begin{cases} \min(1 - \mathbf{x}_0, \epsilon_1) & : \mathbf{z}^k - 1/\rho\mathbf{v}^k > \min(1 - \mathbf{x}_0, \epsilon_1) \\ \max(-\mathbf{x}_0, -\epsilon_1) & : \mathbf{z}^k - 1/\rho\mathbf{v}^k < \max(-\mathbf{x}_0, -\epsilon_1) \\ \mathbf{z}^k - 1/\rho\mathbf{v}^k & : \textit{otherwise} \end{cases}$$

**Updating $\mathbf{z}$:**

$$\mathbf{z}^{k+1} = \arg\min_{\mathbf{z}} \mathcal{L}(\mathbf{x}_0 + \mathbf{z}) - \mathbf{u}^{k^\top}\mathbf{z} - \mathbf{v}^{k^\top}\mathbf{z} + \frac{\rho}{2}\left(\|\eta^{k+1} - \mathbf{z}\|_2^2 + \|\mathbf{w}^{k+1} - \mathbf{z}\|_2^2\right).$$

Liu et al. (2019) showed that the linearized ADMM converges for some non-convex problems. Therefore, by linearizing $\mathcal{L}$ and adding Bergman divergence term $\eta^k/2\|\mathbf{z} - \mathbf{z}^k\|_2^2$, we can then update $z$ as follows:

$$\mathbf{z}^{k+1} = \frac{1}{\eta^k + 2\rho}\left(\eta^k\mathbf{z}^k + \rho(\eta^{k+1} + \frac{1}{\rho}\mathbf{u}^k + \mathbf{w}^{k+1} + \frac{1}{\rho}\mathbf{v}^k) - \nabla\mathcal{L}(\mathbf{z}^k + \mathbf{x}_0)\right).$$

It is worthy to mention that the analysis until this step is inspired by Xu et al. (2018) with modifications to adapt our new formulation.

**Updating $\xi_{\mathbf{A}}$:**

$$\xi_{\mathbf{A}}^{k+1} = \arg\min_{\xi_{\mathbf{A}}} \|\xi_{\mathbf{A}_1}\|_F^2 + \lambda\|\xi_{\mathbf{A}_1}\mathbf{x}_0 - \mathbf{A}_1\eta\|_2^2 + \bar{\mathcal{L}}(\mathbf{A}_1) \ \text{ s.t. } \ \|\xi_{\mathbf{A}_1}\|_{\infty,\infty} \leq \epsilon_2.$$

The previous problem can be solved with proximal gradient methods.

**Experimental Setup.** For the tropical adversarial attacks experiments, there are five different hyper parameters which are

- $\epsilon_1$ : The upper bound for the infinite norm of $\delta$.
- $\epsilon_2$ : The upper bound for the $\|.\|_{\infty,\infty}$ of the perturbation on the first linear layer.
- $\lambda$ : Regularizer to enforce the equality between input perturbation and first layer perturbation
- $\eta$ : Bergman divergence constant.
- $\rho$ : ADMM constant.

**Algorithm 1:** Solving Problem (9)

**Input:** $\mathbf{A}_1 \in \mathbb{R}^{p \times n}, \mathbf{B} \in \mathbb{R}^{k \times p}, \mathbf{x}_0 \in \mathbb{R}^n, t, \lambda > 0, \gamma > 1, K > 0, \xi_{\mathbf{A}_1} = \mathbf{0}_{p \times n}, \eta^1 = \mathbf{z}^1 = \mathbf{w}^1 = \mathbf{z}^1 = \mathbf{u}^1 = \mathbf{w}^1 = \mathbf{0}_n$.

**Output:** $\eta, \xi_{\mathbf{A}_1}$

**Initialize:** $\rho = \rho_0$

**while** not converged **do**

    **for** k $\leq$ K **do**

        $\eta$ **update:** $\eta^{k+1} = (2\lambda \mathbf{A}_1^\top \mathbf{A}_1 + (2 + \rho)\mathbf{I})^{-1}(2\lambda \mathbf{A}_1^\top \xi_{\mathbf{A}_1}^k \mathbf{x}_0 + \rho \mathbf{z}^k - \mathbf{u}^k)$

        **w update:** $\mathbf{w}^{k+1} = \begin{cases} \min(1 - \mathbf{x}_0, \epsilon_1) & : \mathbf{z}^k - 1/\rho \mathbf{v}^k > \min(1 - \mathbf{x}_0, \epsilon_1) \\ \max(-\mathbf{x}_0, -\epsilon_1) & : \mathbf{z}^k - 1/\rho \mathbf{v}^k < \max(-\mathbf{x}_0, -\epsilon_1) \\ \mathbf{z}^k - 1/\rho \mathbf{v}^k & : \textit{otherwise} \end{cases}$

        **z update:** $\mathbf{z}^{k+1} = \frac{1}{\eta^{k+1}+2\rho}(\eta^{k+1}\mathbf{z}^k + \rho(\eta^{k+1} + 1/\rho \mathbf{u}^k + \mathbf{w}^k + 1/\rho \mathbf{v}^k) - \nabla \mathcal{L}(\mathbf{z}^k + \mathbf{x}_0))$

        $\xi_{\mathbf{A}_1}$ **update:**

        $\xi_{\mathbf{A}_1}^{k+1} = \arg\min_{\xi_{\mathbf{A}}} \|\xi_{\mathbf{A}_1}\|_F^2 + \lambda \|\xi_{\mathbf{A}_1}\mathbf{x}_0 - \mathbf{A}_1 \eta^{k+1}\|_2^2 + \bar{\mathcal{L}}(\mathbf{A}_1)$ s.t. $\|\xi_{\mathbf{A}_1}\|_{\infty,\infty} \leq \epsilon_2$

        **u update:** $\mathbf{u}^{k+1} = \mathbf{u}^k + \rho(\eta^{k+1} - \mathbf{z}^{k+1})$

        **v update:** $\mathbf{v}^{k+1} = \mathbf{v}^k + \rho(\mathbf{w}^{k+1} - \mathbf{z}^{k+1}))$

        $\rho \leftarrow \gamma\rho$

    $\lambda \leftarrow \gamma\lambda$

    $\rho \leftarrow \rho_0$

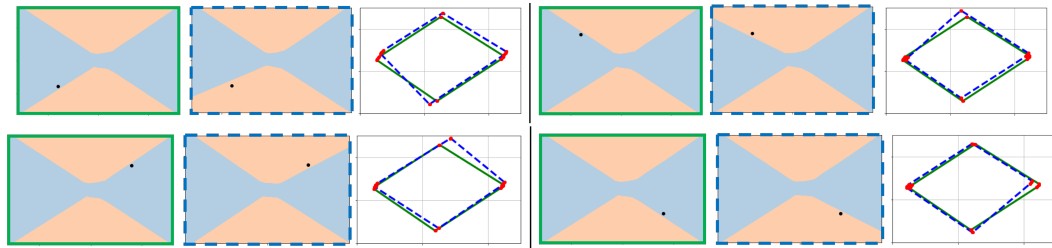

Figure 7: **Dual View of Tropical Adversarial Attacks.** We show the effects of tropical adversarial attacks on a synthetic binary dataset at two different input points (in black). From left to right: the decision regions of the original and perturbed models, and decision boundaries polytopes (green for original and blue for perturbed).

For all of the experiments, we set the values of $\epsilon_2, \lambda, \eta$ and $\rho$ to $1, 10^{-3}, 2.5$ and $1$, respectively. As for $\epsilon_1$ it is set to $0.1$ upon attacking MNIST images of digit 4 set to $0.2$ for all other MNIST images.

**Motivational Insight to the Dual View.** We train a network with 2 inputs, 50 hidden nodes and 2 outputs on a synthetic dataset where we then then solve Equation 9 for a given $\mathbf{x}_0$ shown in black in Figure 7. We show the decision boundaries with and without the perturbation $\xi_{\mathbf{A}_1}$ at the first linear layer. As show in Figure 7, perturbing an edge of the dual subdivision polytope, by perturbing the first linear layer, corresponds to perturbing the decision boundaries and results in the misclassification of $\mathbf{x}_0$. As expected, perturbing different decision boundaries corresponds to perturbing different edges of the dual subdivision. Note that the generated input perturbation $\eta$ is sufficient as well into fooling the network in classifying $\mathbf{x}_0 + \eta$, and by construction is equivalent to perturb the decision boundaries of the network. We show later another example where we alternate the position of $\mathbf{x}_0$ and construct successful adversaries in both the input space, and the parameter space. Furthermore, we conduct experiments on MNIST images in a later section, which show that successful adversarial attacks $\eta$ can be designed by solving Problem (9). Figure 7 shows another example where the sampled to be attacked is closer to a different decision boundary. Observe how the edge corresponding to that decision boundary of the decision boundary polytope has respectively been altered.

**MNIST Experiments.** Here, we design perturbations to misclassify MNIST images. Figure 8 shows several adversarial examples that change the network prediction for digits 8 and 9 to digits 7, 5, and 4, respectively. In some cases, the perturbation $\eta$ is as small as $\epsilon = 0.1$, where $\mathbf{x}_0 \in [0, 1]^n$. Several other adversarial results are reported in Figure 9. We again emphasize that our approach is

not meant to be compared with (or beat) state of the art adversarial attacks but rather to provide a novel geometrically inspired perspective that can shed new light in this field.

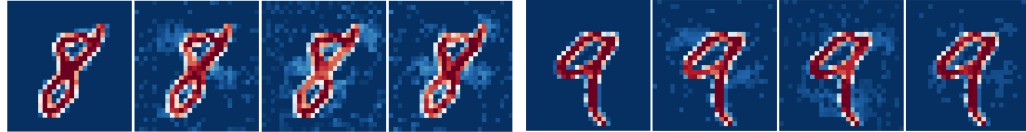

Figure 8: **Effect of Tropical Adversarial Attacks on MNIST Dataset.** We show qualitative examples of adversarial attacks, produced by solving Problem (9), on two digits (8,9) from MNIST. From left to right, images are classified as [8,7,5,4] and [9,7,5,4] respectively.

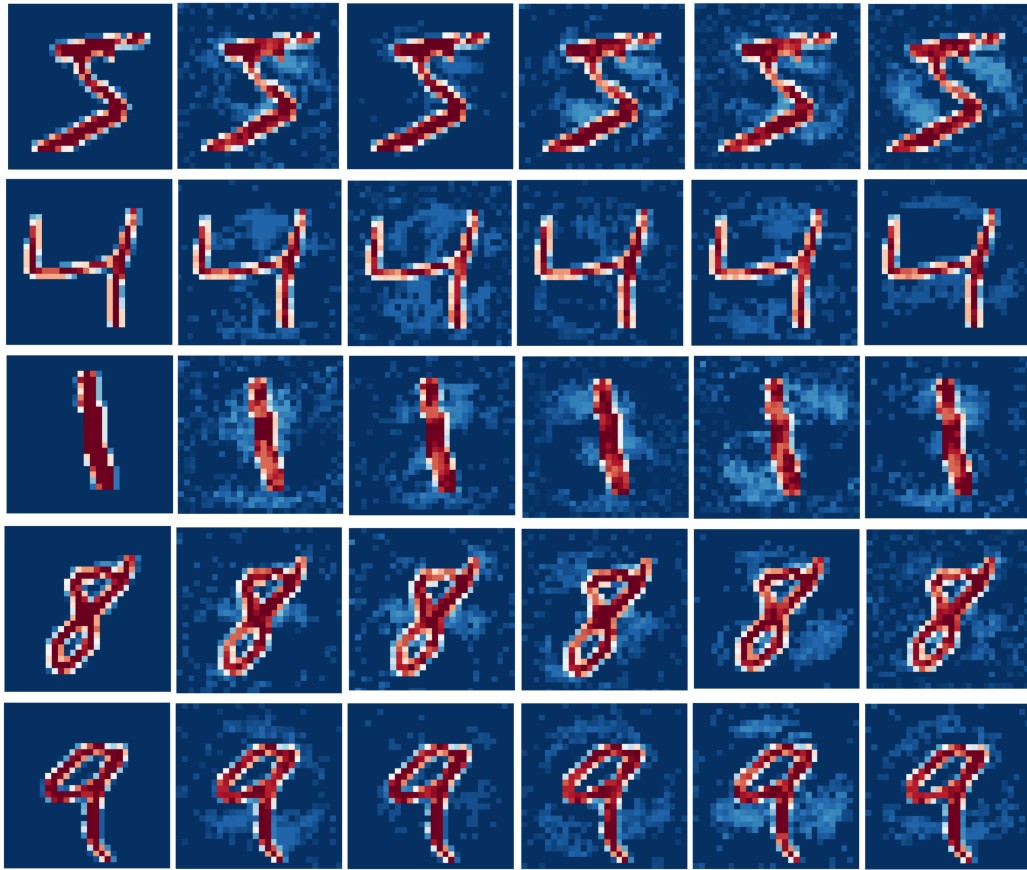

Figure 9: **Effect of Tropical Adversarial Attacks on MNIST Images.** First row from the left: Clean image, perturbed images classified as [7,3,2,1,0] respectively. Second row from left: Clean image, perturbed images classified as [9,8,7,3,2] respectively. Third row from left: Clean image, perturbed images classified as [9,8,7,5,3] respectively. Fourth row from left: Clean image, perturbed images classified as [9,4,3,2,1] respectively. Fifth row from left: Clean image, perturbed images classified as [8,4,3,2,1] respectively.

# 14  EXPERIMENTAL DETAILS AND SUPPLEMENTAL RESULTS

In this section, we describe the settings and the values of the hyper parameters used in the experiments. Moreover, we will show some further supplemental results to the results in the main manuscript paper.

### 14.1 TROPICAL VIEW TO THE LOTTERY TICKET HYPOTHESIS.

We first conduct some further supplemental experiments to those conducted in Section 4. In particular, we conduct further experiments re-affirming the lottery ticket hypothesis on three more synthetic datasets in a similar experimental setup to the one shown in Figure 3. The new supplemental experiments are shown in Figure 10. A similar conclusion is present where the lottery ticket initialization consistently better preserves the decision boundaries polytope compare to other initialization schemes over different percentages of pruning.

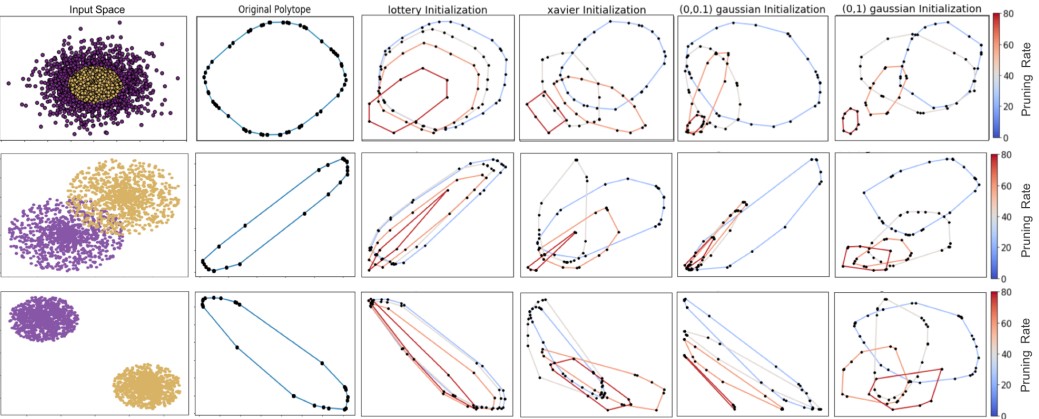

Figure 10: **Effect of Different Initializations on the Decision Boundaries Polytope.** From left to right: training dataset, decision boundaries polytope of original network followed by the decision boundaries polytope during several iterations of pruning with different initializations.

A natural question is whether it is necessary to visualize the dual subdivision polytope of the decision boundaries, *i.e.* $\delta(R(\mathbf{x}))$, where $R(\mathbf{x}) = H_1(\mathbf{x}) \odot Q_2(\mathbf{x}) \oplus H_2(\mathbf{x}) \odot Q_1(\mathbf{x})$ as opposed to visualizing the tropical polynomials $\delta(H_{\{1,2\}}(\mathbf{x}))$ and $\delta(Q_{\{1,2\}}(\mathbf{x}))$ directly for the tropical re-affirmation of the lottery ticket hypothesis. That is similar to asking whether it is necessary to visualize and study the decision boundaries polytope $\delta(R(\mathbf{x}))$ as compared to the the dual subdivision polytope of the functional form of the network since for the 2-output neural network described in Theorem 2 we have that $f_1(\mathbf{x}) = H_1(\mathbf{x}) \oslash Q_1(\mathbf{x})$ and $f_2(\mathbf{x}) = H_2(\mathbf{x}) \oslash Q_2(\mathbf{x})$. We demonstrate this with an experiment that demonstrates the differences between these two views. For this purpose, we train a single hidden layer neural network on the same dataset shown in Figure 3. We perform several iterations of pruning in a similar fashion to Section 5 and visualise at each iteration both the decision boundaries polytope and all the dual subdivisions of the aforementioned tropical polynomials representing the functional form of the network, *i.e.* $\delta(H_{\{1,2\}}(\mathbf{x}))$ and $\delta(Q_{\{1,2\}}(\mathbf{x}))$. It is to be observed from Figure 11 that despite that the decision boundaries were barely affected with the lottery ticket pruning, the zonotopes representing the functional form of the network endure large variations. That is to say, investigating the dual subdivisions describing the functional form of the networks through the four zonotopes $\delta(H_{\{1,2\}}(\mathbf{x}))$ and $\delta(Q_{\{1,2\}}(\mathbf{x}))$ is not indicative enough to the behaviour of the decision boundaries.

### 14.2 TROPICAL PRUNING

**Toy Setup.** To verify our theoretical work, we first start by pruning small networks that are in the form of Affine followed by ReLU followed by another Affine layer. We train the aforementioned network on two 2D datasets with a varying number of hidden nodes (100, 200, 300). In this setup, we observe from Figure that when Theorem 2 assumptions hold, our proposed tropical pruning is indeed competitive, and in many cases outperforms, the other non decision boundaries aware pruning schemes.

**Experimental Setup.** In all experiments of the tropical pruning section, all algorithms are run for only a single iteration where $\lambda$ increases linearly from 0.02 with a factor of 0.01. Increasing $\lambda$ corresponds to increasing weight sparsity and we keep doing until sparsification is 100%.

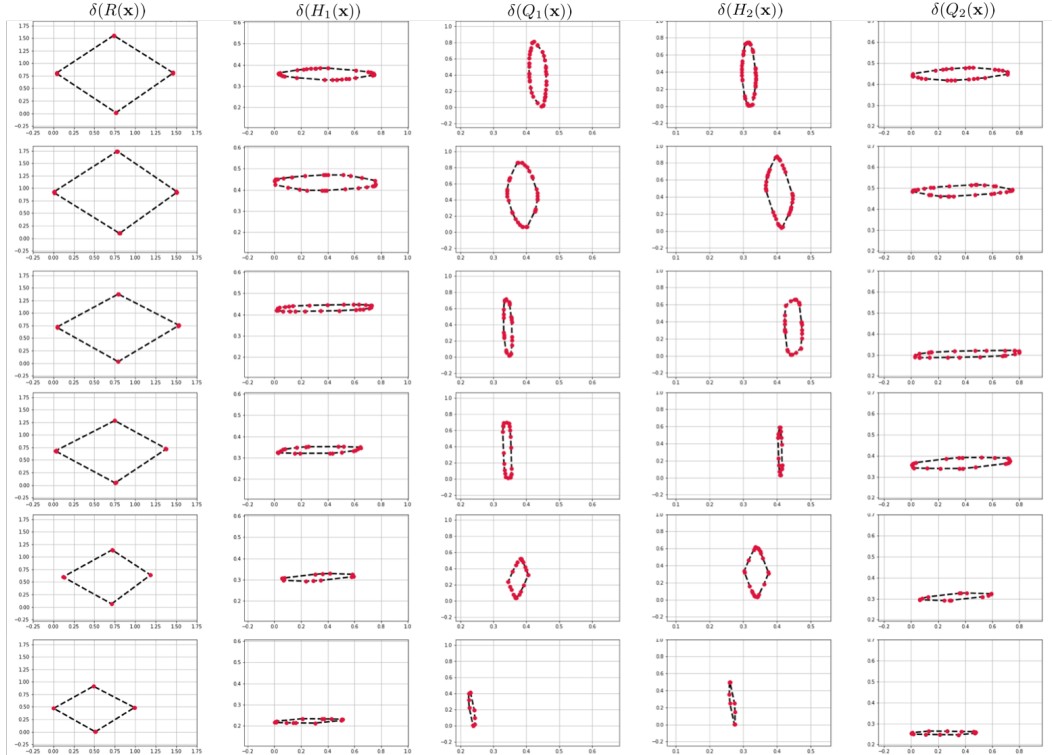

Figure 11: **Comparison between the decision boundaries polytope and the polytopes representing the functional representation of the network.** First column: decision boundaries polytope $\delta(R(\mathbf{x}))$ while the remainder of the columns are the zonotopes $\delta(H_1(\mathbf{x}))$, $\delta(Q_1(\mathbf{x}))$, $\delta(H_2(\mathbf{x}))$ and $\delta(Q_2(\mathbf{x}))$ respectively. Under varying pruning rate across the rows, it is to be observed that the changes that affected the dual subdivisions of the functional representations are far smaller compared to the decision boundaries polytope.

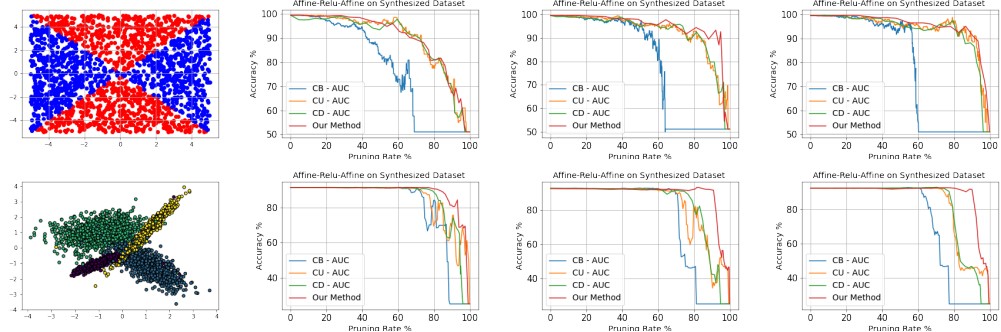

Figure 12: **Pruning Ressults on Toy Networks.** We apply tropical pruning on the toy network that is in the form of Affine followed by a ReLU followed by another Affine. From left to right: (a) dataset used for training (b) pruning networks with 100 hidden nodes (c) 200 hidden nodes (d) 300 hidden nodes.

**Supplemental Experiments.** We conduct more experimental results on AlexNet and VGG16 on SVHN, CIFAR10 and CIFAR100 datasets. We examine the performance for when the networks have only the biases of the classifier fine tuned after tuning as shown in Figure 13. Moreover, a similar experiments is reported for the same networks but for when the biases for the complete networks are fine tuned as in Figure 14.

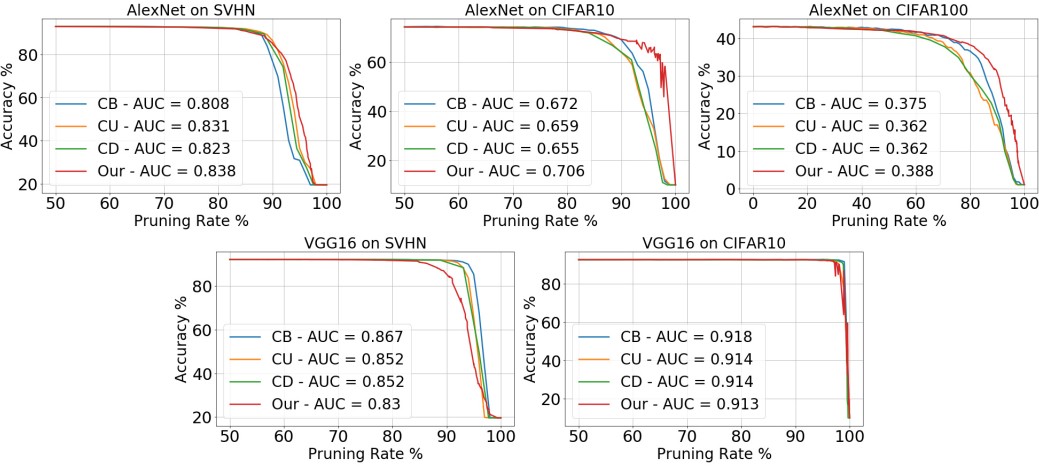

Figure 13: **Results of Tropical Pruning with Fine Tuning the Biases of the Classifier.** Tropical pruning applied on AlexNet and VGG16 trained on SVHN, CIFAR10, CIFAR100 against different pruning methods with fine tuning the biases of the classifier only.

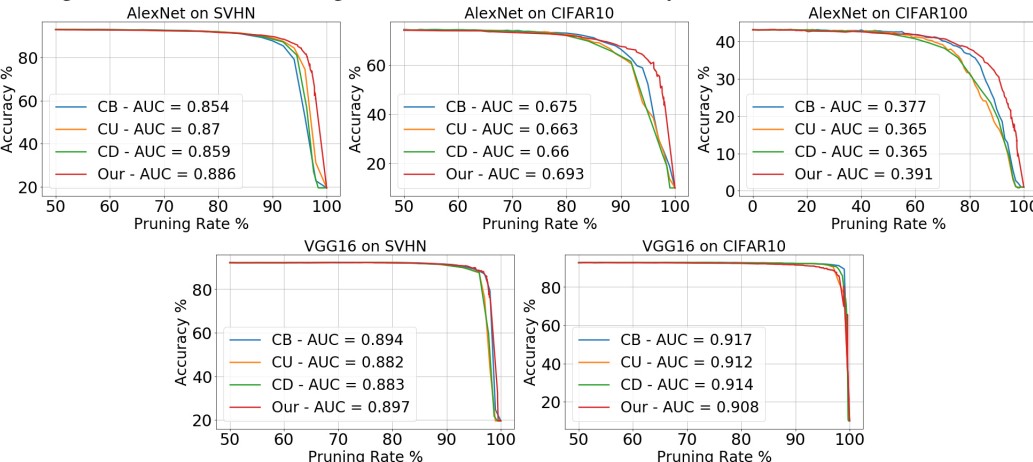

Figure 14: **Results of Tropical Pruning with Fine Tuning the Biases of the Network.** Tropical pruning applied on AlexNet and VGG16 trained on SVHN, CIFAR10, CIFAR100 against different pruning methods with fine tuning the biases of the network.

