# OpenReview forum: "On the Decision Boundaries of Neural Networks. A Tropical Geometry Perspective"
_ICLR.cc/2021/Conference — Reject_

### Official Review · AnonReviewer2 · 2020-10-29
**Outer approximates decision boundary of shallow ReLU network with tropical hypersurface dual to convex hull of two zonotopes.  Applies to various tasks and real data sets, notably network pruning using interesting optimization to preserve the tropical hypersurface.  Tropical geometry content seems more about language.**

**Rating:** 6
**Confidence:** 4

**Review:**

This paper shows the decision boundary of a shallow ReLU network is a subset of a tropical hypersurface, and this tropical hypersurface is dual to a convex hull of two zonotopes given by the weights of the network.  The paper then considers various learning tasks and real data sets, most notably neural network pruning by using an interesting optimization to preserve the tropical hypersurface.  The tropical geometry content seems mostly about language to me.  In particular, the containment of the decision boundary in the tropical hypersurface (first part of Theorem 2) is quite evident, once notation is unpacked.  Also Proposition 1 and Corollary 1 in my opinion can be removed from the document.  On the other hand, the network pruning application that seeks to preserve the convex hull of the zonotopes seems rather substantial to me, with a nice derivation in the appendix.  I would put more emphasis on this myself.

Itemized comments follow.


Page 3: Please be more clear about what type of objects $UF(\mathcal{P}(f))$ and $\delta(f)$ are.  Polyhedral complexes?

Page 3: “For ease, we use ReLUs as the non-linear activation, but any other piecewise linear function can also be used.”  Is it not required for the tropical viewpoint that the slopes of the piecewise linear function be integers?  “Without loss of generality, as one can very well approximate real weights as fractions and multiply by the least common multiple of the denominators as discussed in Zhang et al. (2018).”  Can you remark whether the decision boundaries are stable to such approximations?  A brief discussion on how tropical is using the integral weights assumption may help the reader.

Page 4: typo, “each output can be expressed as a tropical rational as per Theorem 1.”  Tropical rational function

Page 4, Theorem 2: $\delta(R(x)) = ConvHull(\mathcal{Z}_{G_1}, \mathcal{Z}_{G_2})$.  The LHS is a polyhedral complex or subdivision, the RHS looks like just a polytope?  Confusing, until some bit below where it is stated $\delta(R(x))$ actually is not subdivided because there is no bias (so it is just a polytope).

I worry there is only a fine line between “decision boundary” (this paper) and the boundary between the linearity regions of a single output ReLU network (I think early works).  In particular, is $T(R(x))$ also a superset for the boundary between the linearity regions of $(f_1 + f_2)(x)$?

Page 5: “fast algorithms for enumerating these vertices are still restricted to zonotopes with line segments starting at the origin Stinson et al. (2016).”  I find it disingenuous to describe this as a restriction and then to present Proposition 1 as a fix.  This issue is very minor, and I think polytopes people would just say “without loss of generality…”

Pages 7-8: Tropical pruning sounds significantly more computationally intensive than the other simple pruning methods it is being compared to, even when the separability in rows is used.  There also seems to be some awkwardness with the biases since the tropical approach assumes no biases.  While not knowing about pruning of NNs, I still actually found this to be the most substantial and interesting part of the paper and appendix.  It uses the zonotope description of the tropical superset, and involves interesting optimization.  I would suggest the authors emphasize this subsection more.

---

> ### Author Response · Authors · 2020-11-22
> **Response**
>
> **We thank R2 for their comments and review. We hope our following rebuttal is gratifying enough for R2 to raise their score. This is since to our understanding R2's review was mostly clarifying questions and comments. We have updated the pdf with a revised version with the changes marked in red.**
>
>
> - **What type of objects are $\text{UF}(\mathcal{P}(f)$ and $\delta(f)$?** Yes. In general both are polyhedral complexes. This is since that $\delta(f)$ is the set of polytopes each of which is the projection of polytopes forming the upperfaces of $\mathcal{P}(f)$. In the case where the network is bias free, both $\text{UF}(\mathcal{P}(f)$ and $\delta(f)$ are a single polytope which is as per Theorem 2, the convex hull between two zonotopes.
>
>
> - **Integer Weights.** It is theoretically required for the tropical geometry viewpoint that weights are integers. Note that all experiments conducted are on practical networks with real weights and biases. Moreover, one can extend the definition of tropical hypersurfaces for tropical polynomials to tropical signomials (non negative real slopes) where Theorem 2 holds exactly the same. This has been partially addressed in Section 5 Page 7 in \cite{zhang2018tropical}. We have left the exposition of the paper to tropical polynomials to be faithful to the tropical geometry literature.
>
>
> - **Page 4: typo.**  We have addressed the typo in the revised version.
>
>
> - **The LHS is a polyhedral complex or subdivision.** As detailed in our previous comment and our response to R1, $\delta(f)$ is a dual subdivision which is generally a polyhedral complex except in the special case with no biases the dual subdivision will consists of a single polytope.
>
>
> - **I worry there is only a fine line between.** To reiterate, the decision boundaries of a neural network (location where network produce identical prediction scores per class) "happens" to be a subset of the boundaries between linear regions of an artificially constructed tropical polynomial $R(x)$. Throughout the paper, when we say "tropical hypersurfaces are supersets to decision boundaries" we are referring to the decision boundaries of the network when performing classification. The second question by R2 is not clear since each of $f_1$ and $f_2$ are tropical rationals and NOT tropical polynomials.
>
>
> - **Page 5.** We have looked, to the best of our capacity, through the literature and we could not find a similar result. Since proposition 1 is essential to run Algorithm 1 from [2] and to carry the lottery ticket hypothesis experiments, we opted to leave this statement along with the proposition without much elaboration for completeness.
>
>
> - **Pages 7-8.** We thank R2 for their appreciation of our pruning reformulation. While our approach for pruning may sound computationally expensive, it is indeed of similar in complexity to all other pruning methods compared against. This is since the algorithm solving problem (2) in pruning: (a) is separable per row. (b) The per iteration update is closed form and element wise. (c) We conduct a single iteration as we find that this suffices to finding the support of the pruned weights.
>
> [1] "Tropical geometry of deep neural networks"
>
> [2] "A randomized algorithm for enumerating zonotope vertices"

---

### Official Review · AnonReviewer3 · 2020-10-29
**Interesting but incomplete development of applications of tropical geometry to pruning**

**Rating:** 6
**Confidence:** 3

**Review:**

Building upon the observation of Zhang (2018) that a class of deep neural networks have decision boundaries that correspond to tropical rational maps, the paper proposes new methods for pruning networks and constructing adversarial attacks that are based on the idea of minimizing or maximizing the changes to the decision boundary.  The methods focus on optimizing objectives related to the dual polytopes for the networks.

The strength of this paper is in its ambitious agenda: it is difficult to reason about the nonlinear functions learned by neural networks, so the paper’s proposal to use tropical algebra to try to provide new insights is a worthy goal.  Extending beyond Zhang 2018, the paper works out an explicit characterization of the two-layer two-dimensional dual polytopes which is good to see.  The experimental results on pruning of small networks (e.g., Figure 3,Appendix Figure 7) are suggestive that geometric issues may be important for understanding the success of lottery ticket pruning.  The simple observation (beginning with Appendix Eq 9) that adversarial attacks can also be analyzed as attacks on the first layer weights is interesting, although seems like a second separate topic.

While the tropical geometry perspective is welcome, this reviewer found several lines of reasoning insufficiently convincing or unclear in the submitted paper.  The central assertion is that tropical geometry can lead to promising new pruning methods (or at least promising new insights about pruning methods).  However, why the proposed tropical pruning (Section 5) should be expected to work (or why it should be expected to provide insights on the pruning problem) is unclear for several reasons.

1. The tropical pruning method is inspired by the explicit analysis of small networks in the paper, but the method is tested only on deep networks that are beyond the reach of this explicit analysis. This skips an important step in the analysis: is tropical pruning effective on small toy problems?  For example, in the small cases illustrated in Figure 3, tropical pruning is not one of the methods compared.  The paper would be strengthened if the new pruning method could be developed and either explicitly proved or experimentally verified on the small cases which are closer to the settings where the geometry has been explicitly characterized, before exploring claims and experiments in large networks.

2. Why would insights about the geometry of the boundaries be preserved after retraining?  The method reasons about the geometry of the network after it is initially trained but then the paper measures the effect on the network after it is retrained.  The paper would be strengthened if it explained or argued why the lengthy retraining process would be expected to preserve rather than obliterate any geometric properties of the function.

3. The paper does not provide enough intuition and explanation behind the unproved leap from Theorem 2 to Eq 1, and then from Eq 1 to Eq 2.  The text refers to subfigures of Figure 3 (e.g., a first subfigure showing explicit decision boundaries) that seem to be missing.  Figure 4 is a good illustration of the idea, but how the selection of the best change in geometry is related to the proposed objective in Eq 2 is not explained beyond the assertion that “decision boundaries tend to be preserved.”  This assertion should be backed up by more explicit reasoning, more explicit examples or some statistical data.

The work in the appendix bringing in analysis of adversarial examples is interesting but seems to be a second major topic that is not fully developed in the main paper and could be the subject of a different paper.

**edit, based on revision and after discussion**.  Thanks to the authors for answering questions 2 and 3 in discussion and adding an experiment to the paper validating the tropical pruning method in a small setting.  The additional experiment provides some evidence that the tropical pruning method is working as argued, and I change my rating of the paper to 6.

---

> ### Author Response · Authors · 2020-11-23
> **Response**
>
> **We thank R3 for their review. We have updated the pdf with a revised version with the changes marked in red.**
>
> - **On using tropical pruning on small networks.**
> We agree with R3. This is indeed a reasonable experiment that we should have conducted. To that end, we have conducted 6 experiments over two toy datasets on networks with a single hidden layer, satisfying Theorem 2 assumptions, with a varying number of hidden nodes (100, 200, 300). We observe similar conclusions to earlier experiments on deeper networks. Our tropical pruning is either competitive or outperforms other pruning schemes while being on bar with their computational complexity. The new set of experiments can be found in section 14.2 in the \textbf{appendix}.
>
> - **On retraining the network.**
>  The main experimental setup in Section 5 in the paper did **not** perform any retraining. Now that the performance of the pruning inspired by the geometry was established in the main paper, only then and for completeness, we report the experiments for fine tuning the pruned networks in the **appendix**. We do so since that often pruning is a single block within a larger compression pipeline that can involve fine tuning.
>
>  - **The paper does not provide enough intuition.** We respectfully disagree. We will reiterate below the intuition in bullet points to establish the raised questions about the link from Thm (2) to Eq (1) and from Eq (1) to Eq (2).
>
>      - (a) Theorem 2 provides a superset to the decision boundaries. The superset can be characterized through the dual subdivision polytope.
>
>      - (b) As shown in Figure 4, there could be multiple dual subdivisions each corresponding to a different network with the same decision boundaries. Thus, can one construct a dual subdivision that corresponds to a network that is sprase?
>
>      - (c) Approximating a dual subdivision with another (that leads to sparser network) so as to preserve decision boundaries can be difficult; therefore, can one approximate the dual subdivision with the generators of the zonotopes directly?
>
>  The link from Theorem 2 to Eq 1 is through the link between (a) to (b). Theorem 2 provides the motivation for why would one want to approximate a dual subdivision.  The link between Eq 2 and Eq (1) is through the link between (b) and (c). Now that Eq (1) is difficult, while enforcing a sparsely represented network, Eq (2) is to the rescue. This is since instead of directly approximating the dual subdivision, Eq 2 approximates the generators of the zonotopes by another set of generators that are sparser (for pruning).
>
>  - **On the analysis of tropical adversarial attacks.**
>  We believe that breaking this up into separate work will dilute the contributions and may not fit with a similar nice story line where we propose new theory along with several applications. Moreover, there are several accepted papers to top tier conferences that briefly discuss another contribution but leave the detailed explanation to the appendix. See [1] for an ICML example where importance sampling is detailed thoroughly only in the appendix.
>
> [1] "SGD: General Analysis and Improved Rates"

---

### Official Review · AnonReviewer4 · 2020-11-01
**Interesting problem and solution, more experiments are desired**

**Rating:** 6
**Confidence:** 1

**Review:**

This submission proposes the use of tropical geometry to understand the decision boundaries of neural networks. The targeted problem is quite interesting and the proposed method looks to be solid; however, I didn't thoroughly check the correctness of the proposed theorems.

In the experiments on tropical pruning, since for VGG16, the proposed method only shows better performance on the CIFAR10 dataset, more experiments are desired to make a strong conclusion. For instance, performing experiments on ResNet.
Why changing to use LeNet when comparing to Tropical geometry approaches?

How about the efficiency of the proposed approach? Is it practical to use the proposed method in a deep neural network?

A minor issue: In the results section, "Figure 4" should be "Figure 5".

---

> ### Author Response · Authors · 2020-11-22
> **Response**
>
> **We thank R4 for their comments and review. We have updated the pdf with a revised version with the changes marked in red.**
>
>
> - **On ResNets experiments.** Performing experiments on ResNets is interesting, however in this case pruning may not deliver significant gains in terms of memory reduction as ResNets contain only a few linear layers (where most memroy complexity lies) and are memory efficient compared to VGG.
>
> - **On using LeNet in comparison.** We followed the same experimental setup in [1,2] for the sake of fair comparison.
>
>
> - **On the efficiency of proposed pruning approach.** As mentioned in page 7, the optimization is conducted for a single step where the updates are expressed in closed form. This makes the use of our proposed method in deep networks cheap, practical and with similar complexity to all competitors.
>
>  - **On the typo in results** It has been fixed in the revised version.
>
>
> [1] "Multiclass Neural Network Minimization via Tropical Newton Polytope Approximation"
>
> [2] "Tropical polynomial division and neural networks"

---

### Official Review · AnonReviewer1 · 2020-11-03
**Review of "On the Decision Boundaries of Neural Networks. A Tropical Geometry Perspective"**

**Rating:** 7
**Confidence:** 3

**Review:**

Summary:  This work studies the decision boundaries of neural networks (NN) with piecewise linear (ReLU) activation functions from a tropical geometry perspective. Leveraging the work of [1], the authors show that NN decision boundaries form subsets of tropical hypersurfaces. This geometric characterization of NN decision boundaries is then leveraged to better understand the lottery ticket hypothesis, and prune deep NNs. The authors also allude to the use of tropical geometric perspectives on NN decision boundaries for the generation of adversarial samples, but do not explicitly discuss it in any detail within the main text of the paper.

Strengths:
+ The paper is insightful and novel.
+ Tropical geometry (TG) promises to be a particularly convenient language to study (ReLU) DNNs, and this work does a good job of showcasing the versatility afforded by a principled, geometric approach in improving our understanding of DNNs.
+ The efforts put into making this paper accessible to readers unfamiliar with TG are also worth appreciating.

Weaknesses:
- The paper perhaps bites off a little more than it can chew. It might be best if the authors focused on their theoretical contributions in this paper, added more text and intuition about the extensions of their current bias-free NNs, fleshed out their analyses of the lottery ticket hypothesis and stopped at that.

- The exposition and experiments done with tropical pruning need more work. Its extension to convolutional layers is a non-trivial but important aspect that the authors are strongly encouraged to address. This work could possibly be written up into another paper. Similarly, the work done towards generating adversarial samples could definitely do with more detailed explanations and experiments. Probably best left to another paper.

Contributions:  The theoretical contributions of the work are significant and interesting. The fact that the authors have been able to take their framework and apply it to multiple interesting problems in the ML landscape speaks to the promise of their theory and its resultant perspectives. The manner in which the tropical geometric framework is applied to empirical problems however, requires more work.

Readability: The general organization and technical writing of the paper are quite strong, in that concepts are laid out in a manner that make the paper approachable despite the unfamiliarity of the topic for the general ML researcher. The language of the paper however, could do with some improvement; Certain statements are written such that they are not the easiest to follow, and could therefore be misinterpreted.

Detailed comments:
- While there are relatively few works that have explicitly used tropical geometry to study NN decision boundaries, there are others such as [2] which are similar in spirit, and it would be interesting to see exactly how they relate to each other.

- Abstract: It gets a little hard to follow what the authors are trying to say when they talk about how they use the new perspectives provided by the geometric characterizations of the NN decision boundaries. It would be helpful if the tasks were clearly enumerated.

- Introduction: “For instance, and in an attempt to…” Typo – delete “and”. Similar typos found in the rest of the section too, addressing which would improve the readability of the paper a fair bit.

- Preliminaries to tropical geometry: The preliminaries provided by the authors are much appreciated, and it would be incredibly helpful to have a slightly more detailed discussion of the same with some examples in the appendix. To that end, it would be a lot more insightful to discuss ex. 2 in Fig. 1, in addition to ex. 1. What exactly do the authors mean by the “upper faces” of the convex hull? The dual subdivision and projection $\pi$ need to be explained better.

- Decision boundaries of neural networks: The variable ‘p’ is not explicitly defined. This is rather problematic since it has been used extensively throughout the rest of the paper. It would make sense to move def. 6 to the section discussing preliminaries.

- Digesting Thm. 2: This section is much appreciated and greatly improves the accessibility of the paper. It would however be important, to provide some intuition about how one would study decision boundaries when the network is not bias-free, in the main text. In particular, how would the geometry of the dual subdivision $\delta(R({\bf x}))$ change? On a similar note, how do things change in practice when studying deep networks that are not bias free, given that, “Although the number of vertices of a zonotope is polynomial in the number of its generating line segments, fast algorithms for enumerating these vertices are still restricted to zonotopes with line segments starting at the origin”? Can Prop. 1 and Cor. 1 be extended to this case trivially?

- Tropical perspective to the lottery ticket hypothesis: It would be nice to quantify the (dis)similarity in the shape of the decision boundaries polytopes across initializations and pruning using something like the Wasserstein metric.

- Tropical network pruning: How are $\lambda_1, \lambda_2$ chosen? Any experiments conducted to decide on the values of the hyper-parameters should be mentioned in the main text and included in the appendix. To that end, is there an intuitive way to weight the two hyper-parameters relative to each other?

- Extension to deeper networks: Does the order in which the pruning is applied to different layers really make a difference? It would also be interesting to see whether this pruning can be parallelized in some way. A little more discussion and intuition regarding this extension would be much appreciated.

- Experiments:
- The descriptions of the methods used as comparisons are a little confusing – in particular, what do the authors mean when they say “pruning for all parameters for each node in a layer” Wouldn’t these just be the weights in the layer?
- “…we demonstrate experimentally that our approach can outperform all other methods even when all parameters or when only the biases are fine-tuned after pruning” – it is not immediately obvious why one would only want to fine-tune the biases of the network post pruning and a little more intuition on this front might help the reader better appreciate the proposed work and its contributions.
- Additionally, it might be an unfair comparison to make with other methods, since the objective of the tropical geometry-based pruning is preservation of decision boundaries while that of most other methods is agnostic of any other properties of the NN’s representational space.
- Going by the results shown in Fig. 5, it would perhaps be better to say that the tropical pruning method is competitive with other pruning methods, rather than outperforming them (e.g., other methods seem to do better with the VGG16 on SVHN and CIFAR100)
- “Since fully connected layers in DNNs tend to have much higher memory complexity than convolutional layers, we restrict our focus to pruning fully connected layers.” While it is true that fully connected layers tend to have higher memory requirements than convolutional ones, the bulk of the parameters in modern CNNs still belong to convolutional layers. Moreover, the most popular CNNs are now fully convolutional (e.g., ResNet, UNet) which would mean that the proposed methods in their current form would simply not apply to them.
- Comparison against tropical geometry approaches for network pruning – why are the accuracies for the two methods different when 100% of the neurons are kept and the base architecture used is the same? The numbers reported are à (100, 98.6, 98.84)
Tropical adversarial attacks: Given that this topic is not at all elaborated upon in the main text (and none of the figures showcase any relevant results either), it is strongly recommended that the authors either figure out a way to allocate significantly more space to this section, or not include it in this paper. (The idea itself though seems interesting and could perhaps make for another paper in its own right.)

- References:  He et al. 2018a and 2018b seem to be the same.

[1] Zhang L. et al., “Tropical Geometry of Deep Neural Networks”, ICML 2018.
[2] Balestriero R. and Baraniuk R., “A Spline Theory of Deep Networks”, ICML 2018.

---

> ### Author Response · Authors · 2020-11-22
> **Response (1/2)**
>
> **We thank R1 for their detailed response and appreciation for our contribution. We addressed all typos and suggestions by R1 and marked them in red in the updated pdf.**
>
> - **In Regards to the Weaknesses.** We respectfully disagree with R1. While the adversarial attacks section was left for the appendix, it flows better with the current exposition of the paper. Breaking one piece of work into several chunks will break the story flow and dilute contributions. However, we agree that our paper left several other open questions from extensions to convolutional layers to handling biases. Those directions have indeed enough material on their own to stand as a separate future work.
>
> - **Other Similar Works [2].** It seems that R1 missed posting the reference. We will be more than happy to add and elaborate on that once provided.
>
> - **Abstract.** As  suggested, we have refined the abstract and enumerated the tasks.
>
> - **Preliminaries to Tropical Geometry.** Following R1's advise, we have dedicated a full new Section 8 in the appendix that operationalizes the construction of the tropical hypersurface and dual subdivision of the second example tropical polynomial in Figure 1.
>
> - **Decision boundaries of neural networks.** The $p$ in Definition 6 is an arbitrary choice for the generic definition of zonotopes indicating the number of line segments. For better exposition, and as suggested, we have moved Definition 6 to the preliminaries and changed $p$ to $L$.
>
> - **Digesting Thm. 2..** The ease in computing the dual subdivision for a bias free network is due to the following: (1) the upperfaces of the newton polytope $\mathcal{P}(f)$ is $\Delta(f)$. That is one would need to compute the convex hull between the convex hull between the two zonotopes in $\mathbb{R}^n$ to find the upperfaces. (2) The projection $\pi$ is an identify mapping. Thus and as per the definition of the dual subdivision (below Definition 6), $\delta(f) = \{\pi(p) \subset \mathbb{R}^d : p \in \text{UP}(\mathcal{P}(f))\} = \{p : p \in \Delta(f)\}$. Therefore the dual subdivision which is a polyhedral complex $\delta(f)$ is a set of a single polytope. However, once biases are introduced, and while $\mathcal{P}(f)$ which is the convex hull between two zonotopes still holds for Theorem 2,  constructing the upperfaces $\text{UP}(\mathcal{P}(f))$ to construct the dual subdivision (which is a polyhedral complex) $\delta(f)$ is no longer trivial. This entails that while Corollary 1 can be extended to biases trivially by appending the bias dimension, this is only for the construction of the newton polytope and NOT the dual subdivision. As for proposition 1, it holds regardless of the biases as it is for generic zonotopes. From a practical perspective, we find that performing pruning with the bias free assumption, as a proxy, works reasonably well in practice. We encourage R1 to read our newly added Section 2 in our appendix to see a visual example of the construction of the dual subdivision in the presence of the biases. Also section C of [1] provides nice visual examples too.
>
> - **Tropical perspective to the lottery ticket hypothesis.** We will look into Wasserstien distance and the convex geometry literature for other distances that could capture the information about the orientation of polytopes. It is worthy to mention that we have investigated the use of Hausdorff distance in pruning in optimization (1) but we did not find that it gives any particular advantage over approximating the generators directly as in optimization (2).
>
> - **On $\lambda_1$ and $\lambda_2$.** While $\lambda_1$ and $\lambda_2$ control the pruning-approximation trade-off, we find that the optimization solution is not very sensitive to their choices where we set them to be both equal (details of their values is in the appendix). We believe this is since we perform the optimization for only a single iteration to determine the support the support of the weights.
>
> - **Extension to deeper networks.** This is interesting; however, we did not conduct any experiments in this direction. We generally do not believe that the order matters and that if every block on its own preserves its own dual subdivision the overall composition should preserve the decision boundary. How sensitive is the decision boundary to every independent approximation is yet to be investigated.
>
> - **Descriptions of the methods.** R1 is correct. We have refined this sentence.
>
> - **"we demonstrate experiment".** We have addressed this in red in the revised version.

---

> > ### Author Response · Authors · 2020-11-22
> > **Response (2/2)**
> >
> > - **It might be an unfair comparison.** We respectfully disagree. We believe it is fair to compare against other pruning methods, even when ours is aware of decision boundaries, so long our method is of similar order complexity to other methods. This is the case since we only carry the optimization of (2) for a single iteration to determine the support. Let alone we believe that part of progress is to devise smarter algorithms/methods that use more problem structure.
> >
> > - **Going by the results shown in Fig. 5.** We agree with R1. We have revised the statement.
> >
> > - **"Since fully connected layers in DNNs".** The description of R1 is correct. However, we believe that this is a good starting point to future directions. Linking two domains together is not trivial while providing theoretical results that has nontrivial several applications. Moreover, we would like to point to R1 similar work that conducted a much less complex setup, only fully connected small networks on smaller datasets [2], that has been accepted to top tier conferences among many other papers.
> >
> > - **"Comparison against tropical geometry approaches for network pruning ".** Trained models were not provided by [2,3]. Thereafter we report the same results reported in their Tables 1/2. For our approach as mentioned in the paper, we follow the same training pipeline as in [2,3] to perform our pruning. We believe the differences in accuracy at $100\%$ kept neurons is due to different initialization before training.
> >
> > - **Tropical Adversarial Attack.** We believe that breaking this up into separate work will dilute the contributions and may not fit with a similar nice story line where we propose new theory along with several applications.
> >
> > - **"References".** We thank R1 for their keen observation. We have deleted the replicated reference.
> >
> >
> > [1] "Tropical geometry of deep neural networks"
> >
> > [2] "Multiclass Neural Network Minimization via Tropical Newton Polytope Approximation"
> >
> > [3] "Tropical polynomial division and neural networks"

---

### Decision · Program_Chairs · 2021-01-07
**Final Decision**

**Decision:**

Reject

**Comment:**

This paper studies the decision boundaries of shallow ReLU network using the formalism of tropical geometry. Its main takeaway is to provide a new interpretation of the lottery ticket hypothesis in terms of network pruning strategies that preserve certain geometric structure.

Reviewers were appreciative of the clarity of the exposition, and the novel perspective on interesting and elusive phenomena such as the lottery ticket hypothesis. On the other hand, they also expressed some doubts about the significance of some aspects of the theory (such as proposition 1 and corollary 1), as well as the computational considerations required to elevate the analysis to large-scale architectures from applications.

Ultimately, and after taking into consideration all the reviewing discussions, the AC believes that this submission is not yet ready for publications, but it is in a trajectory to become an important piece of work. In particular, the AC encourages delving deeper into the tropical network pruning. Additionally, the authors might want to discuss [Breaking the Curse of Dimensionality with Convex Neural Networks, Bach'17] in the related work, since this is the first instance the AC is aware of where the connection between zonotopes and shallow ReLU networks is established.